# A mass spectrum-oriented computational method for ion mobility-resolved untargeted metabolomics

Mingdu Luo[1,2], Yandong Yin[1], Zhiwei Zhou [1], Haosong Zhang[1,2], Xi Chen[1,2], Hongmiao Wang[1,2] & Zheng-Jiang Zhu [1,3] ✉

Ion mobility (IM) adds a new dimension to liquid chromatography-mass spectrometry-based untargeted metabolomics which significantly enhances coverage, sensitivity, and resolving power for analyzing the metabolome, particularly metabolite isomers. However, the high dimensionality of IM-resolved metabolomics data presents a great challenge to data processing, restricting its widespread applications. Here, we develop a mass spectrum-oriented bottom-up assembly algorithm for IM-resolved metabolomics that utilizes mass spectra to assemble four-dimensional peaks in a reverse order of multidimensional separation. We further develop the end-to-end computational framework Met4DX for peak detection, quantification and identification of metabolites in IM-resolved metabolomics. Benchmarking and validation of Met4DX demonstrates superior performance compared to existing tools with regard to coverage, sensitivity, peak fidelity and quantification precision. Importantly, Met4DX successfully detects and differentiates co-eluted metabolite isomers with small differences in the chromatographic and IM dimensions. Together, Met4DX advances metabolite discovery in biological organisms by deciphering the complex 4D metabolomics data.

Mass spectrometry-based untargeted metabolomics has become increasingly successful in providing a comprehensive, unbiased and quantitative characterization of metabolites in a biological system under investigation[1,2]. However, the enormous chemical and the compositional diversity of metabolome presents a grand challenge in separation and identification of metabolites. This challenge is particularly prominent for isomeric metabolites, which tend to co-elute on chromatography and share similar fragments in MS/MS spectra[3,4]. Recently, liquid chromatography–ion mobility–mass spectrometry (LC–IM–MS) has been emerging as a promising technology for four-dimensional untargeted metabolomics[5,6]. Ion mobility adds a new dimension of separation to LC–MS. After ionization, ion mobility separates metabolite ions according to their sizes, shapes, and charges in gas phase on the millisecond timescale, which effectively improves

the resolving power and selectivity without adding to analysis times[7–9]. The ion mobility-derived collision cross-section (CCS) is a valuable physiochemical property for metabolite identification with high reproducibility across different instruments[10–13]. The hyphenation of IM with LC and tandem MS provides a promising strategy for the multidimensional analysis of the complex metabolome[13–16]. This technology has demonstrated distinct advantages over conventional LC–MS, including improved separation and peak capacity, capability to separate and differentiate metabolite isomers, and generation of four-dimensional (4D) data (i.e., $m/z$, retention time, CCS, and MS/MS spectrum) for metabolite identification[6,17].

The high complexity of ion mobility-resolved untargeted metabolomics presents a great challenge to data processing. In particular, peak detection becomes more challenging in ion mobility-based four-

[1]Interdisciplinary Research Center on Biology and Chemistry, Shanghai Institute of Organic Chemistry, Chinese Academy of Sciences, Shanghai 200032, P. R. China. [2]University of Chinese Academy of Sciences, Beijing 100049, P. R. China. [3]Shanghai Key Laboratory of Aging Studies, Shanghai 201210, P. R. China. ✉e-mail: jiangzhu@sioc.ac.cn

dimensional data, which restricts the broad application of 4D metabolomics. Compared to increasing software availability in ion mobility-enhanced 4D proteomics[18–21], very few software tools were developed for 4D metabolomics[22–24]. Mass spectra generated from ion mobility-resolved untargeted metabolomics are structurally complex because of the additional ion mobility dimension. For 4D peak detection in such complex dataset, the top-down based dimensionality reduction is often employed. For example, in 4D proteomics, software tools such as MaxQuant[18] and IonQuant[19] slice the 4D data space ($m/z$, retention time, ion mobility, and intensity) into multiple 3D sub-spaces ($m/z$, retention time, and intensity) along the ion mobility dimension (Supplementary Fig. 1a). Then, peak detection is performed in each 3D sub-space using the conventional peak detection algorithm in LC–MS. This strategy finally tracks along the ion mobility dimension and integrates the same $m/z$-RT peaks in multiple 3D sub-spaces to form a 4D peak. Alternatively, in 4D lipidomics, MS-DIAL[22] developed a top-down compressing strategy for dimensionality reduction. Specifically, the 4D data space was compressed into one 3D space ($m/z$, retention time, and intensity) by summing the ion mobility dimension (Supplementary Fig. 1b). Peak detection is first performed in retention time dimension. For each detected LC peak, a second peak detection is applied in ion mobility dimension. In summary, current 4D peak detection methods aim to convert 4D data into 3D data through the top-down based dimensionality reduction. Although dimensionality reduction simplifies the data structures, it also introduces signal masking and artifacts, and may reduce the peak detection sensitivity. To foster a more widespread application of 4D untargeted metabolomics and accelerate its development, more informatic solutions are urgently required.

The LC–IM–MS empowered 4D metabolomics enables to sequentially separate metabolites according to the required timescales from LC, IM to MS (Fig. 1a). Inspired by the sequential separations, in this work, we developed a mass spectrum-oriented bottom-up assembly algorithm for 4D peak detection. The bottom-up assembly strategy considers a mass spectrum (MS1 or MS2) as the smallest unit in 4D dataset, and builds its related elution peaks in IM dimension and LC dimension, respectively, in a reverse order of multidimensional separation. We refer to this as a reverse engineering strategy from data acquisition. With our 4D peak detection algorithm, we further developed an end-to-end computational framework, namely Met4DX, for peak detection, quantification and identification of metabolites in 4D untargeted metabolomics (Fig. 1b–e). We demonstrated that Met4DX substantially improved 4D peak detection coverage and sensitivity in different biological samples. Performances of Met4DX were thoroughly benchmarked and validated with other existing tools, including coverage, sensitivity, and peak fidelity as well as quantification precision. Importantly, Met4DX enabled to detect and differentiate co-eluted isomeric metabolites even with small differences on LC and IM separations. Finally, we demonstrated that Met4DX supported accurate identification of metabolites, in particular for co-eluted metabolites. In summary, Met4DX is a mass spectrum-oriented end-to-end computational tool for peak detection, quantification, and identification of metabolites in 4D untargeted metabolomics, which will empower the widespread applications of ion mobility-resolved metabolomics.

## Results

### The Met4DX workflow

Met4DX provides an end-to-end computational workflow for peak detection, quantification, and metabolite annotation in ion mobility-enhanced 4D untargeted metabolomics (Fig. 1 and Supplementary Fig. 2). A MS spectrum is the smallest unit in 4D data acquisition, while the MS2 spectrum is highly essential for metabolite annotation. Thus, Met4DX starts the 4D data processing from each individual MS2 spectrum, and includes 4 major modules: (1) MS2 spectral dereplication; (2) the bottom-up assembly algorithm for 4D peak detection;

(3) 4D peak alignment and grouping; and (4) multidimensional match for metabolite annotation.

In data-dependent MS2 acquisition (DDA), for example, parallel accumulation serial fragmentation (PASEF)-DDA in trapped ion mobility mass spectrometer (TIMS)[25], the same precursor ion is usually chosen and fragmented multiple times to generate redundant MS2 spectra. To address this issue, Met4DX first performs MS2 spectral dereplication (Fig. 1b). Specifically, all acquired MS2 spectra are binned using the 3D information of their precursor ions (MS1), including $m/z$, retention time, and ion mobility (see Methods). For each bin, 3D distances between any of two MS2 spectra are calculated by integrating the similarities of ion mobility and RT of their precursor ions and MS2 spectra. Then, a hierarchical cluster analysis (HCA) is applied using the 3D distances among MS2 spectra in the bin, and generates the unique MS2 clusters for each bin. Finally, the MS2 spectrum with the highest spectral intensity (termed as unique MS2 spectrum) is selected to represent the cluster. The dereplication step significantly reduces the redundancy of MS2 spectra for subsequent peak detection.

Second, we developed a MS spectrum-oriented bottom-up assembly algorithm for 4D peak detection, which is the core module of Met4DX (Fig. 1c). The 4D peak detection is performed via 5 steps starting from individual MS2 spectrum. In step 1, for each unique MS2 spectrum, Met4DX searches its precursor MS1 data point in the $m/z$-ion mobility data frame with the precursor $m/z$, ion mobility, and frame index recorded in the mgf file. In step 2, the adjacent MS1 data points from the same $m/z$-ion mobility frame are retrieved to assemble a possible ion mobility peak. Then, peak detection is performed in ion mobility dimension. If a successful extracted ion mobilogram (EIM) peak is obtained, in step 3, MS1 data retrieval and assembly of EIM peaks are repeatedly extended to the adjacent $m/z$-ion mobility frames (i.e., EIM extension). After that, in step 4, all MS1 data points from EIM peaks in each mobility frame are summed up to rebuild the ion chromatogram in LC dimension. Similarly, peak detection is also performed in the assembled ion chromatogram (i.e., EIC detection). If a successful EIC peak is obtained, in step 5, a 4D peak is constructed with recorded apex and boundary information in EIM and EIC detections. Peak intensity is generated from 3D peak volume. Most importantly, Met4DX employs a serial of criteria during 4D peak detection to reduce false positive and improve peak fidelity (see Methods). Collectively, peak detection in Met4DX starts from one MS2 spectrum and constructs the 4D peak in a reverse order of the sequential separation in LC–IM–MS. Therefore, we refer our peak detection as a bottom-up assembly strategy.

Third, 4D peak alignment and grouping is performed to combine multiple samples for quantification (Fig. 1c). Specifically, metabolic peaks are matched between samples using 4D information ($m/z$, retention time, mobility, and MS/MS spectrum). Matched peaks are employed as landmarks to construct a retention time regression model for retention time correction and peak alignment. A density-based peak grouping strategy is further applied with the distribution of $m/z$ and ion mobility values. Peak match between samples is also performed to overcome the stochasticity during data acquisition. Finally, gap filling is applied for missing values through targeted extraction. A 4D peak table is generated for metabolite quantification in multiple samples and groups, while related 4D information for each feature ($m/z$, retention time, CCS, and MS/MS spectrum) is also outputted. Fourth, metabolite annotation is performed through multidimensional match (Fig. 1d). In Met4DX, we curated a metabolite library of 135,638 metabolites collected from KEGG and HMDB with calculated $m/z$, CCS, RT, and MS/MS spectra (experimentally acquired or in silico predicted). In summary, Met4DX is a MS spectrum-oriented computational tool to provide an end-to-end data processing for 4D untargeted metabolomics. Met4DX is now freely available as an R package in GitHub.

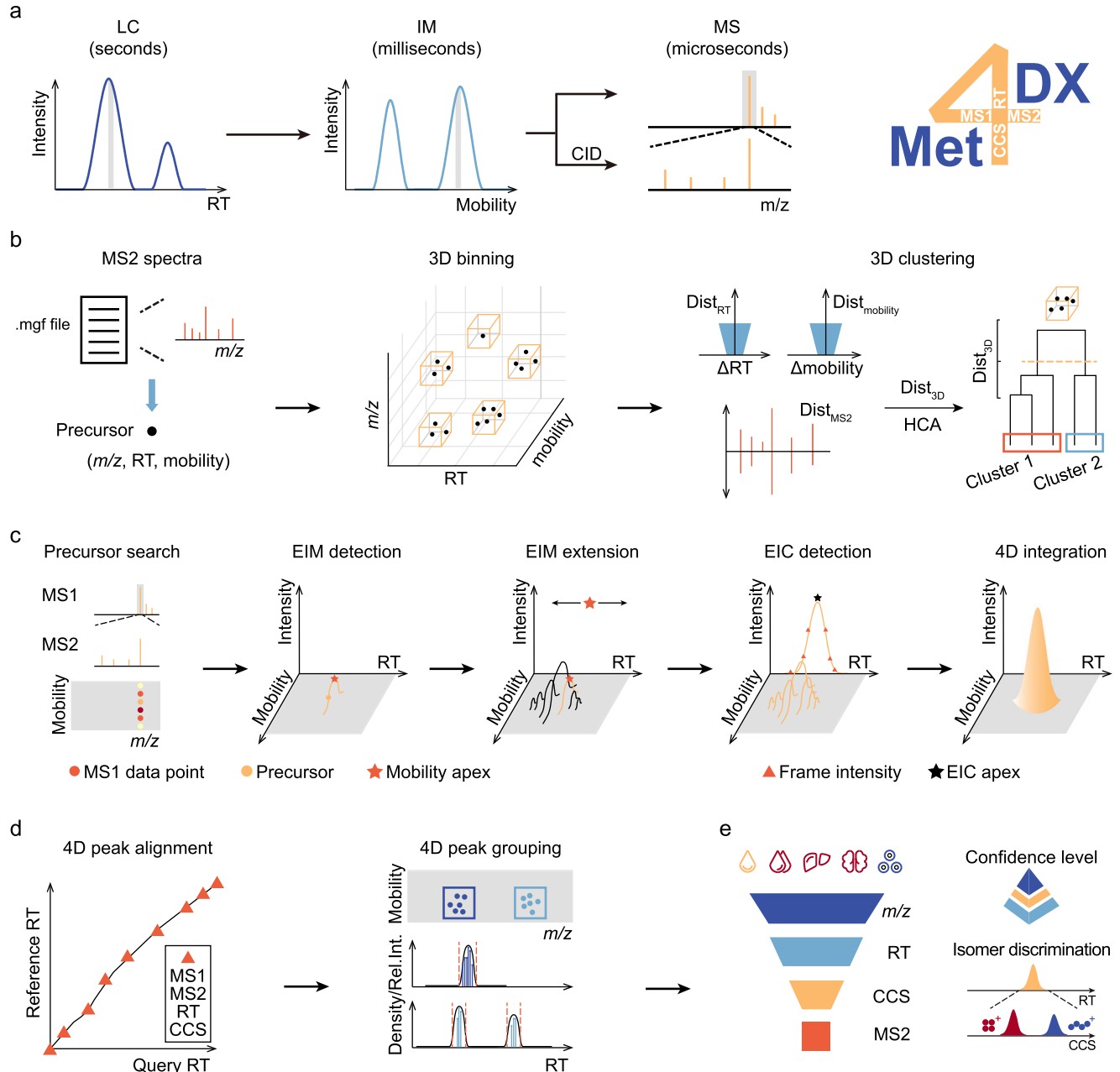

**Fig. 1 | The Met4DX workflow for ion mobility-resolved untargeted metabolomics. a** The multidimensional separations and measurements of metabolites using LC–IM–MS/MS technology. **b–e** Four major modules in Met4DX: **b** MS2 spectral dereplication; **c** the bottom-up assembly algorithm for 4D peak detection; **d** 4D peak alignment and grouping; and **e** multidimensional match for metabolite annotation. The black dot in (**b**) represents the precursor ion of an MS2 spectrum.

## 4D peak detection

We demonstrated 4D peak detection of Met4DX with NIST human urine samples (n = 6 technical replicates) acquired using TIMS with PASEF-DDA technology. In positive mode, a total of ~25,000 MS2 spectra acquired from each sample were inputted into Met4DX for 3D binning and MS2 spectral dereplication (Fig. 2a). Most clusters had only one unique MS2 spectrum (~60%; Supplementary Fig. 3). A total of ~8600 unique MS2 spectra were generated for each replicate (64% reduction; Fig. 2b). With them as inputs, the MS2 spectrum-oriented bottom-up assembly was performed for 4D peak detection in each sample, and ~6700 features were successfully detected (Fig. 2c). On average, 78% of unique MS2 spectra enabled to achieve the successful de novo assembles of 4D features. Finally, the application of peak alignment and grouping generated a total of 7287 features with a

minimal fraction of 0.5 applied (Fig. 2d). The negative mode data is shown in Supplementary Fig. 4. In negative mode, as high as 98% of unique MS2 spectra enabled to achieve the successful de novo assembles of 4D features.

More specifically, an example of kynurenic acid was employed to demonstrated the bottom-up assembly algorithm for 4D peak detection in Mex4DX (Fig. 2e). After MS2 dereplication, a unique MS spectrum (m/z = 190.0500 Da; mobility, $1/K_0$ = 0.640 V·s/cm²; frame index = 1056) was generated to initiate peak detection. Met4DX first searched and retrieved its precursor data point with m/z and mobility information in the frame 1506. Then, adjacent MS1 data points in the same frame were also retrieved to assemble the IM peak. The peak apex was determined at $1/K_0$ of 0.640 V·s/cm². The same IM peak assembly was repeatedly implemented in adjacent frames (frame index 966 – 1146;

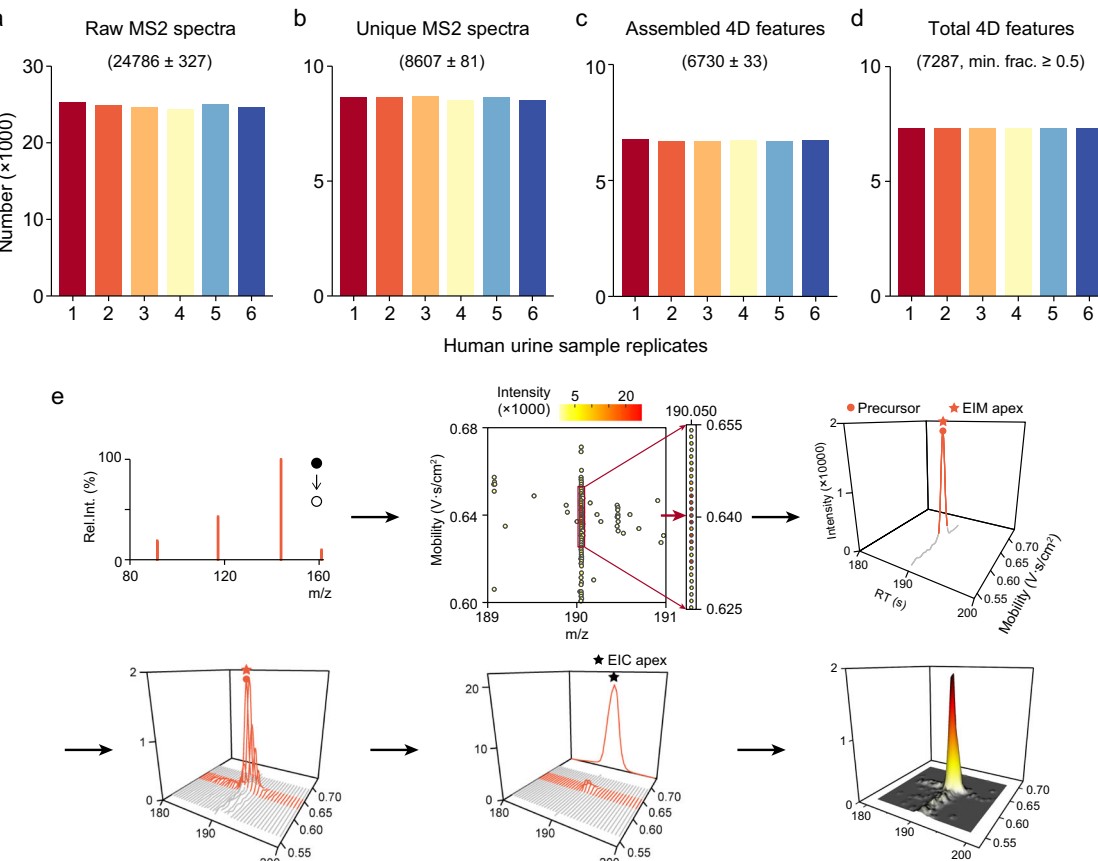

**Fig. 2 | 4D peak detection in human urine samples.** NIST human urine samples were acquired using TIMS with PASEF-DDA technology in positive mode and processed using Met4DX (*n* = 6 technical replicates). **a**–**c** The numbers of raw MS2 spectra (**a**), unique MS2 spectra (**b**), and assembled 4D features (**c**) in each sample replicate. Numbers represent mean ± SD (*n* = 6 technical replicates). **d** The total number of 4D features with a minimal fraction of 0.5 applied. **e** An example of kynurenic acid in urine sample was employed to demonstrate the bottom-up assembly algorithm for 4D peak detection in Mex4DX. Red arrow in the *m/z*-ion mobility data frame indicates the precursor data point for the unique MS2 spectrum. The combined symbols of open/closed cycles and the arrow in the unique MS2 spectrum represent a product ion scan to generate an MS2 spectrum from a fixed precursor ion.

Fig. 2e). All IM peaks in these frames were summed up to assemble the corresponding EIC peak in LC dimension, and the EIC peak apex was determined as 190 s. Finally, peak intensity was integrated from the peak volume. This 4D peak was identified as kynurenic acid with four-dimensional match (*m/z*, CCS, RT, and MS/MS) and validation by the chemical standard (Supplementary Fig. 5).

## Performance benchmark

The MS spectrum-oriented bottom-up assembly strategy in Met4DX enables 4D peak detection with high coverage and sensitivity. Here, we further employed five different types of biological samples (NIST human urine, NIST human plasma, mouse brain tissue, mouse liver tissue, and 293T cells) to evaluate peak detection performance of Met4DX, including coverage, sensitivity, and peak fidelity. Other software tools such as MS-DIAL and MetaboScape (Bruker Daltonics, Bremen, Germany) were used for comparison. Met4DX achieved a total of 6500–8700 and 4400–6800 features in positive and negative modes, respectively, for different sample types (Fig. 3a, b, Supplementary Data 1–3). The benchmark analyses demonstrated that Met4DX significantly improved peak coverage by 2–3 folds compared to MS-DIAL and MetaboScape. In human urine samples, we demonstrated that ~70% features reported in MS-DIAL or MetaboScape were also found in Met4DX (Fig. 3c, d; see Methods). The similar results were also obtained in other samples (Supplementary Fig. 6). To clarify, we only counted 4D peaks with MS2 spectra obtained from MS-DIAL and MetaboScape for benchmark. To demonstrate high sensitivity, we serially diluted the human urine samples by 10 to 1000 times (10X,

100X, and 1000X), and processed the data with different software tools (Fig. 3e, f). Met4DX provided significantly higher sensitivity of peak detection in all diluted samples. In particular, Met4DX enabled to detect a total of 2951 peaks in 1000X samples. As a comparison, only ~600 peaks were detected from the same samples using MS-DIAL and MetaboScape (Fig. 3e). Thus, in low abundant samples, Met4DX significantly increased the peak detection coverage by ~5 folds. The similar results were also obtained for negative mode (Fig. 3f).

We further verified the peak fidelity of 4D peaks obtained from Met4DX through both manual check and the deep learning-empowered software EVA developed by Huan group[26]. A 4D peak with good EIC or EIM peak shape was recognized as a true 4D peak. We selected the replicate sample with the highest peak intensity for evaluating peak fidelity. For NIST human urine samples, we exported and manually checked 7287 EIC and 7287 EIM peaks in positive mode, and 6889 EIC and 6889 EIM peaks in negative mode, respectively. For NIST human urine samples, 97% and 93% of 4D peaks were manually checked with good peak fidelity in positive and negative modes, respectively (Fig. 3g). Also, EVA checked the same samples and reported the similar peak fidelity rates of 96% and 95% in positive and negative mode, respectively. In addition, 93% and 89% of peak fidelity evaluations were consistency between manual check and EVA in positive and negative modes, respectively (Supplementary Fig. 7). Moreover, EVA was used to evaluate the peak fidelity of all biological samples, and showed that Met4DX achieved high rates of peak fidelity (92% ± 5%) in all biological samples (Fig. 3g, Supplementary Data 4). MS-DIAL and MetaboScape also achieved similarly high rates of peak

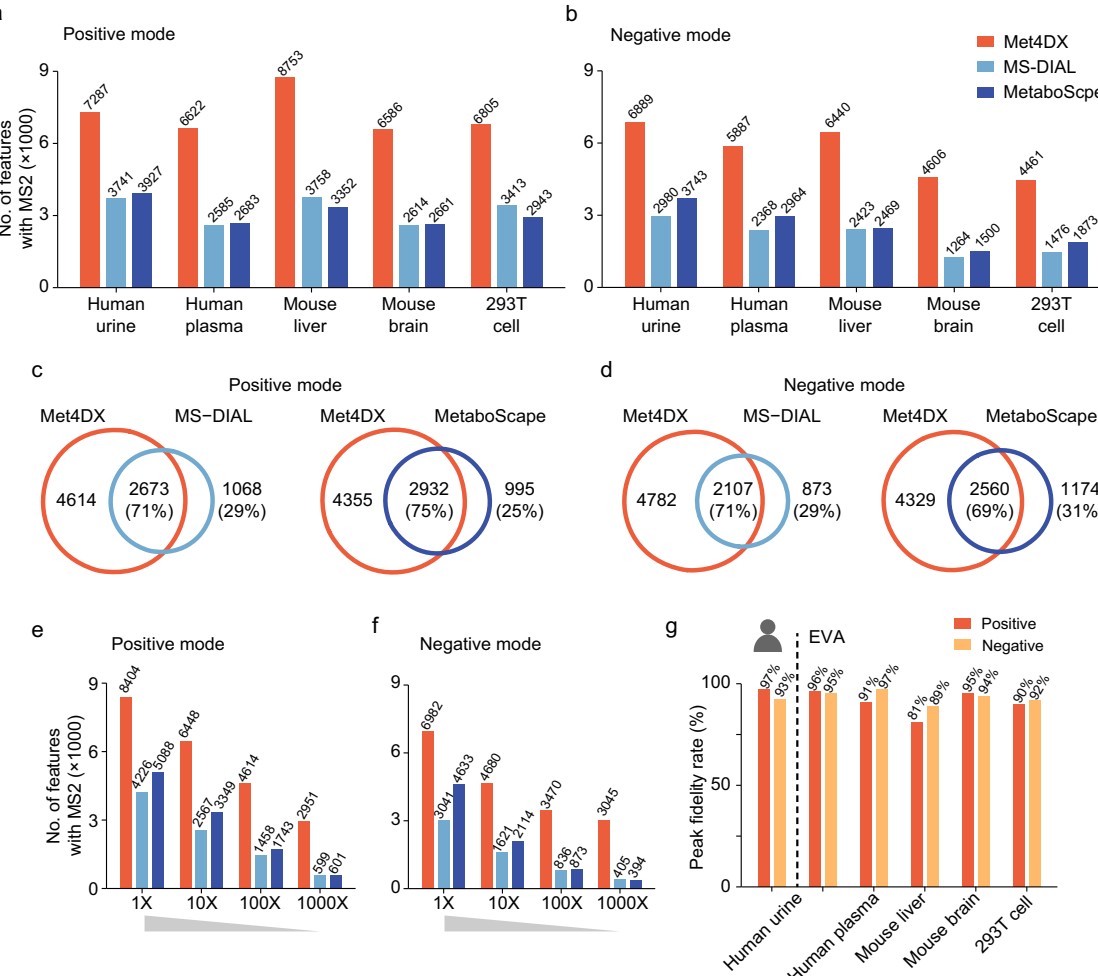

**Fig. 3 | High coverage and sensitivity of 4D peak detection. a, b** The numbers of 4D peaks with MS2 spectra achieved by different software tools in different biological samples in positive (**a**) and negative (**b**) modes (*n* = 6 technical replicates for each sample type). **c, d** Overlaps of 4D peaks in human urine samples detected by different software tools in positive (**c**) and negative (**d**) modes. **e, f** The numbers of detected 4D peaks in serially diluted urine samples (10 to 1000 times) in positive (**e**) and negative (**f**) modes (*n* = 6 technical replicates for each dilution sample). **g** Peak fidelity rates of 4D peaks in biological samples obtained from Met4DX. In **a**–**f**, we only counted 4D peaks with MS2 spectra obtained from MS-DIAL and MetaboScape for the comparisons.

fidelity (97% ± 3% and 96% ± 3%, respectively; Supplementary Fig. 8 and Supplementary Data 5 and 6). For computational resources comparison, we run these software tools with NIST human urine data (positive mode) on a desktop with Intel Core i7-12700 (2.10 GHz; 12 CPU cores; 20 logical cores) and 32 GB memory. For computational time, MS-DIAL, MetaboScape, and Met4DX took 3, 6, and 20 min to finish the data processing. For memory usage, MS-DIAL, MetaboScape, and Met4DX occupied ~4–6 GB, ~1–1.5 GB, and ~6–9 GB, respectively. In general, the computing resource required by Met4DX is affordable for most users. Collectively, we demonstrated that Met4DX enables a high-coverage 4D peak detection through the bottom-up assembly strategy, and achieves peak detection with high sensitivity and peak fidelity.

**Quantification precision**

To evaluate the quantification precision of Met4DX, we ran it, as well as MS-DIAL and MetaboScape on 6 technical replicates of human urine samples. The distribution of relative standard deviations (RSDs) of peak intensities indicates that Met4DX had substantially higher quantification precision compared to MS-DIAL and MetaboScape (Fig. 4a–c), with median RSDs of 16.7%, 22.0%, and 17.9%, respectively. Also, percentages of detected peaks with RSD less 30% were 81%, 61%, and 67% for Met4DX, MS-DIAL, and MetaboScape, respectively.

Exemplified scatter plots of 4D feature intensities between two replicates were displayed for Met4DX and MS-DIAL (Fig. 4d, e). Met4DX clearly demonstrated fewer outliers and higher Pearson correlation than MS-DIAL. All pairwise Pearson correlations between replicates were demonstrated as a heat map for both Met4DX and MS-DIAL (Fig. 4f), showing consistently higher correlations for Met4DX (median 0.974) compared to MS-DIAL (median 0.919). The similar results were also obtained in negative mode data of human urine samples (Supplementary Fig. 9). However, we found that MetaboScape had significantly lower pairwise Pearson correlations (median 0.305) due to the high portions of missing values (Supplementary Fig. 10). Furthermore, we evaluated the quantitation linearity of Met4DX. A mixture of 20 natural products was spiked into NIST human urine samples in a 4-fold dilution series. All of natural products measured by LC–IM–MS and detected by Met4DX showed good quantitation linearity (see Methods, Supplementary Fig. 11 and Supplementary Data 7).

**Metabolite annotation and isomer differentiation**

Met4DX is an end-to-end computational tool for 4D untargeted metabolomics, which provides metabolite annotation through multidimensional match with the metabolite library. In Met4DX, we curated a metabolite library of 135,638 metabolites collected from KEGG and HMDB with calculated *m/z*, CCS, RT, and MS/MS spectra (experimental

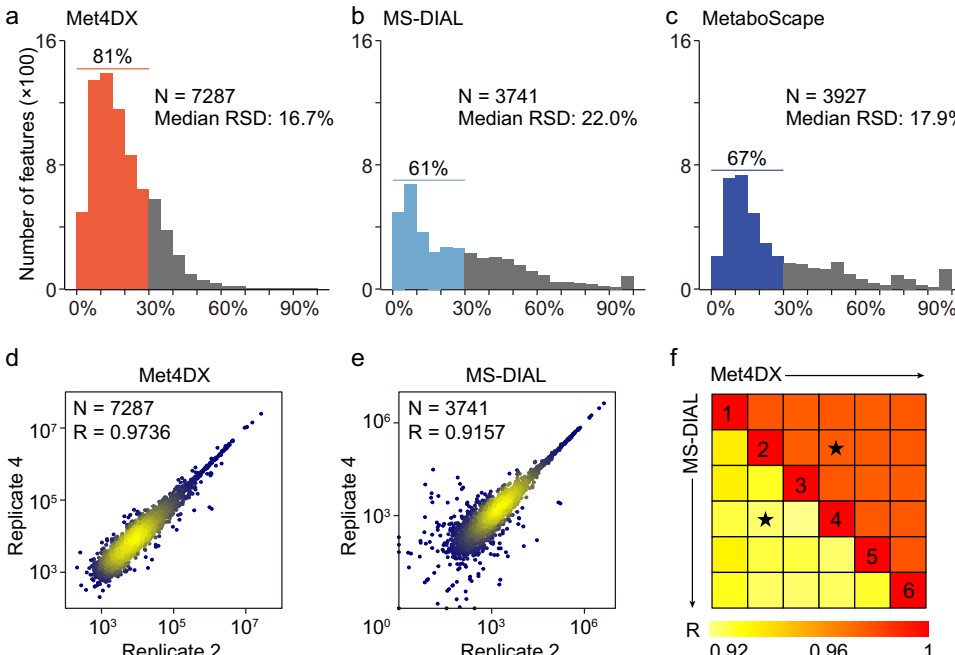

**Fig. 4 | High quantification precision of 4D peak detection. a–c** The distributions of relative standard deviations (RSDs) of 4D peaks from human urine samples detected by Met4DX (**a**), MS-DIAL (**b**), and MetaboScape (**c**). **d, e** Log-log scatter plot of peak intensities between sample replicates 2 and 4 obtained from Met4DX (**d**) and MS-DIAL (**e**) for human urine samples. The color represents the peak density, in which yellow means dense and blue means sparse. **f** All pairwise Pearson correlations of peak intensities in 6 replicates of human urine samples using Met4DX (upper triangle) and MS-DAL (lower triangle). The urine samples were measured in positive mode (*n* = 6 technical replicates).

or in silico predicted; see Methods and Supplementary Fig. 12a). According to the definitions of Metabolomics Standards Initiative (MSI)[27], we classified the confidence levels of metabolite annotations in Met4DX as follows: level 1, 4D match with MS1, CCS, experimental RT, and experimental MS2 spectra from in-house chemical standards; level 2, 4D match with MS1, CCS, predicted RT, and experimental MS2 spectra from public libraries; level 3, 4D match with MS1, CCS, predicted RT, and predicted MS2 spectra (Fig. 5a and Supplementary Fig. 12b). With features from Met4DX, we annotated 339 and 219 features with level 1 annotations, 163 and 55 features with level 2 annotations, and 1636 and 1309 features with level 3 annotations in mouse liver samples in positive and negative modes, respectively (Fig. 5b). The ion mobility-enhanced metabolite annotations of other biological samples were provided in Supplementary Fig. 13 and Supplementary Data 8.

Isomeric metabolites are common in biological samples, and tend to co-elute on chromatography and share similar fragments in MS/MS spectra[3,4]. The ion mobility-enhanced untargeted metabolomics is particularly prominent for differentiating isomeric metabolites. In our metabolite library, 125,173 out of 135,638 metabolites (92%) have at least one isomer (with the same formula). A total of 1,648,4071 pairs of metabolite isomers were generated. We calculated RT and CCS values for all metabolites and found that LC could resolve 18% of isomer pairs (ΔRT ≥ 10 s; Fig. 5c). The addition of IM significantly improved the resolving power, and 39% of isomer pairs could be fully resolved and separated on baseline by IM (ΔCCS ≥ 4%; Fig. 5c). Most modern commercial IM-MS instruments provide an IM resolving power (ca. 50–100, CCS/ΔCCS) that separates metabolite isomers with a half-peak-width when their ΔCCS was larger than 2%[28,29]. With this resolution, 62% of isomer pairs could be differentiated with LC×IM dual separations (ΔCCS ≥ 2%). For isomeric pairs with 0.5% of ΔCCS, the separation can only be achieved when the IM resolving power reaches >250[30]. In this case, 92% of isomeric pairs could be resolved (Fig. 5c). We also demonstrated how LC separation differentiated isomeric metabolites

with different resolutions in the metabolite library (Supplementary Fig. 14). By comparison, LC showed relatively poorer separation than IM. However, the combination of LC and IM further improved the differentiation of isomeric metabolites.

We further evaluated whether Met4DX enabled to sensitively detect and differentiate the isomeric metabolites in real 4D metabolomics data. For example, a total of 6440 4D features were detected by Met4DX in negative mode of mouse liver samples. Among them, a total of 3033 pairs of co-eluted isobaric features were successfully detected and differentiated by Met4DX (Δ*m/z* ≤ 10 ppm and ΔRT ≤ 10 s; Fig. 5d). As a comparison, MS-DIAL and MetaboScape only detected 827 and 1262 pairs, respectively (Fig. 5d). Similar results were also obtained in positive mode of mouse liver samples (Supplementary Fig. 15). Therefore, Met4DX increased the coverage of co-eluted isobaric features by 2–4 folds. The ΔCCS distribution of co-eluted isobaric features detected by Met4DX were demonstrated in Fig. 5e. Among them, 1654 pairs (54%) had ΔCCS larger than 4%, which were fully separated by IM at baseline. In addition, 513 and 180 pairs had ΔCCS in the ranges of 2–4% and 1–2%, respectively. An example pair of M206T169C147 and M206T166C155 discriminated by Met4DX was shown in Fig. 5f. This pair of co-eluting isobaric feature pairs had ΔCCS of 4.8%, fully separated by ion mobility at baseline. The detailed bottom-up assembly based 4D peak detection was provided in Supplementary Fig. 16. With a smaller ΔCCS of 2.6% and separation at half-peak-height by IM, feature pairs of M692T122C236 and M692T122C242 were also discriminated by Met4DX (Fig. 5g and Supplementary Fig. 17). Another isobaric feature pair with IM separation at half-peak-height and ΔCCS of 2.3% in positive mode were shown in Fig. 5h and Supplementary Fig. 18. As bottom-up assembly peak detection avoided dimensionality reduction, Met4DX enabled to differentiate small CCS difference and had high sensitivity on isobaric feature discrimination. Therefore, Met4DX also enabled to discriminate co-eluted pairs of isobaric features with ΔCCS as small as 1–2% (Fig. 5i and Supplementary Fig. 19), which could only be partially separated by IM and better resolved with

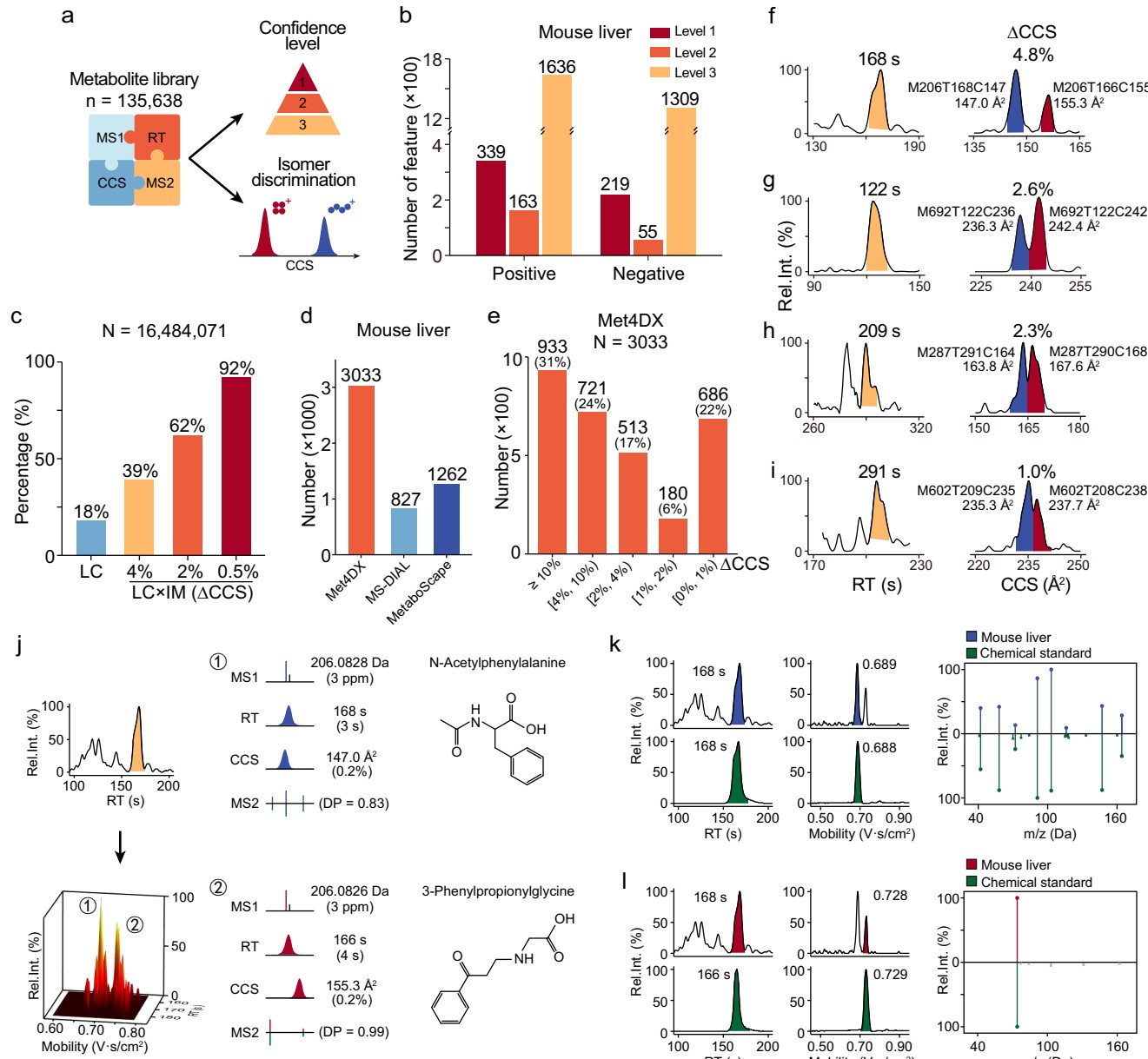

**Fig. 5 | Ion mobility-enhanced multidimensional metabolite annotation and isomer differentiation. a** The curation of 4D metabolite library in Met4DX to support metabolite annotation. **b** Numbers of annotated 4D features with different confidence levels in positive and negative modes of mouse liver samples ($n = 6$ technical replicates). **c** Differentiation of isomeric metabolites with the same formula in 4D metabolite library with LC separation and LC×IM dual separations. For LC separation, ΔRT was set as ≥10 s. For IM separation, ΔCCS were set as ≥4%, ≥2%, and ≥0.5%, respectively. In our metabolite library, 125,163 of 135,638 metabolites had at least one isomeric metabolite (with the same formula), which generated a

total of 16,484,071 pairs of isomeric metabolite pairs. **d** The pair number of co-eluted isobaric features detected in different software tools. We defined the isobaric feature pairs had Δm/z ≤ 10 ppm and ΔRT ≤ 10 s in each software tool. **e** The distributions of ΔCCS for co-eluted feature pairs in mouse liver samples. **f–i** Examples of co-eluted feature pairs with different ΔCCS detected by Met4DX but missed by MS-DIAL and MetaboScape (Supplementary Fig. 21). **j** Identifications of metabolite isomers in panel (**f**) were achieved by Met4DX. **k, l** Validations of N-acetyl-L-phenylalanine (**k**) and 3-phenylpropionylglycine (**l**) by chemical standards, respectively.

LC×IM dual separation in LC–IM–MS (Supplementary Fig. 20). As comparison, MS-DIAL and MetaboScape failed to detected these isobaric feature pairs in Fig. 5f–i (Supplementary Fig. 21), which proved the superiority of Met4DX for differentiation of isomeric metabolites.

Interestingly, as many as 686 pairs of isobaric features with ΔCCS less than 1% also successfully detected by Met4DX. The marginal ΔCCS indicated that they could not be resolved by IM. We calculated ΔRT of these isobaric feature pairs, and demonstrated that they were resolved by LC×IM dual separations and successfully detected by Met4DX. For example, the isobaric feature pairs of M765T377C255 and M765T383C255 in Supplementary Fig. 22a had the same CCS values.

Met4DX discriminated them in the step of EIC detection during the bottom-up assembly based 4D peak detection. Another example of co-eluted isobaric feature pairs of M1023T495C265 and M1023T497C267 were also shown in Supplementary Fig. 22b. With ΔCCS of 0.7% and ΔRT of 1.1 s, this pair were also successfully discriminated by Met4DX.

In multidimensional metabolomic analyses using LC–IM–MS, LC separation also played a vital role in separation of isomeric metabolites in real biological samples. For example, in the mouse liver samples, there were 3691 features with at least one IM co-eluting isobaric feature (Δm/z ≤ 10 ppm and ΔCCS ≤ 2%), generating a total of 14,528 IM co-eluting isobaric feature pairs. Among them, 12,690 pairs (87%) had

ΔRT larger than 20 s (Supplementary Fig. 23). These results proved the importance of LC separation in multidimensional metabolomics using LC–IM–MS.

The detection of co-eluted isobaric feature pairs by Met4DX facilitated accurate annotation of metabolites, in particular for isomer metabolites. The co-eluted isobaric features in Fig. 5f, M206T169C147 and M206T166C155 were identified as N-acetyl-L-phenylalanine and 3-phenylpropionylglycine, respectively, through matching 4D metabolite library by Met4DX (Fig. 5j). The co-eluted isobaric feature pairs were annotated as the isomeric metabolites of N-acetyl-L-phenylalanine and 3-phenylpropionylglycine, respectively, with the sequential $m/z$, RT, CCS, and MS2 spectral matches. Among them, the CCS match successfully differentiated them (Supplementary Fig. 24). The identifications were further validated by chemical standards (Fig. 5k, l). As a comparison, the same biological sample was also acquired on LC-MS, and a chimeric MS2 spectrum was obtained (Supplementary Fig. 25). Although the chimeric MS spectrum included major fragments from both metabolites, the 3D library match only generated the identification of 3-phenylpropionylglycine and failed to identify N-acetyl-L-phenylalanine. As MS-DIAL and MetaboScape failed to detect this pair of isobaric features (Supplementary Fig. 21a, e), the isomers were not identified by them.

Together, we demonstrated that Met4DX enabled to sensitively detect and differentiate the isomeric metabolites in real 4D metabolomics data and increased the coverage by 2–4 folds. Most importantly, due to the bottom-up assembly algorithm developed for 4D peak detection, co-eluted isomeric metabolites even with small differences on LC and IM separations could be successfully differentiated by Met4DX. We have demonstrated that Met4DX is an end-to-end computational tool for 4D untargeted metabolomics, which provides IM-enhanced metabolite annotation and is particularly prominent for differentiating isomeric metabolites.

## Versatile data processing in Met4DX

Met4DX is a versatile tool to process various types of IM-based 4D metabolomics data. In addition to process PASEF-DDA data starting from MS2 spectra, Met4DX also enables to process data with a user-inputted list of precursor ions to initiate the bottom-up assembly 4D peak detection (Supplementary Figs. 26 and 27), which further increases the coverage of peak detection, in particular for those without MS2 spectra acquired. To facilitate this workflow, we have curated a list of precursor ions collected from various biological samples reported above ($N = 72,265$ in positive mode; $N = 42,553$ in negative mode; Supplementary Data 9). Each ion includes $m/z$, RT, and CCS information. With the inputted precursor ion list, Met4DX enables to perform 4D peak detection and related data processing for different IM-MS data including PASEF-DDA and PASEF-DIA data from Bruker TIMS, and IM-AIF data from Agilent DTIM-MS. For example, in NIST human urine samples acquired by PASEF-DDA, Met4DX detected 11419 features in total. Among them, 8309 features (73%) had MS2 spectra (Supplementary Fig. 28a, Supplementary Data 10). As a comparison, MS-DIAL and MetaboScape detected 14440 and 4351 features, and 3741 (26%) and 3927 (90%) features had MS2 spectra, respectively. The percentage of 4D features with MS2 spectra in MS-DIAL was significantly lower than Met4DX and MetaboScape. Moreover, Met4DX also showed higher quantification precision with a median RSD of 16.5%. The percentage of detected peaks with RSD less 30% was as high as 75% in Met4DX (Supplementary Fig. 28b). As a sharp contrast, although MS-DIAL generated more 4D features, most of them had poor quantification precision (median RSD = 47.8%; Supplementary Fig. 28c). Moreover, we overlapped 4D features detected by Met4DX and MS-DIAL (Supplementary Fig. 29a). As a result, 5263 features were obtained from both software tools, while 6156 and 9711 features were obtained only from Met4DX and MS-DIAL, respectively. However, these additional features obtained from MS-DIAL had low

quantification precision with a median RSD of 56.3% and percentage of peaks with RSD less than 30% was as low as 11% (Supplementary Fig. 29b). In addition, only 11% of 4D peaks had MS2 spectra acquired (Supplementary Fig. 29d). These results indicated that additional peaks detected by MS-DIAL and missed by Met4DX had low quality. As a sharp comparison, additional features from Met4DX showed much higher quantitation precision and MS2 coverage (Supplementary Fig. 29c, e).

We also demonstrated the capability of Met4DX to process PASEF-DIA data from Bruker TIMS (Supplementary Fig. 30, Supplementary Data 11) and AIF data from Agilent DTIM-MS instruments (Supplementary Fig. 31, Supplementary Data 12). In NIST human urine data acquired by PASEF-DIA from TIMS, Met4DX detected 10021 features. Among them, 7882 features (79%) had MS2 spectra extracted (Supplementary Fig. 30a). As a comparison, for the same dataset, MS-DIAL detected 10681 features, but only 104 features had MS2 spectra extracted (<1%, Supplementary Fig. 30b). MetaboScape detected 5753 features and did not support MS2 spectral extraction for DIA data (Supplementary Fig. 30c). Together, these results demonstrated that Me4DX enabled high-coverage 4D peak detection and MS2 spectral extraction in PASEF-DIA data compared with other software tools. Both MS-DIAL and MetaboScape have to improve MS2 spectral extraction function for PASEF-DIA data. For IM-AIF data from Agilent DTIM-MS, Met4DX detected 8324 features in NIST human urine data, and 8318 features had MS2 spectra extracted (Supplementary Fig. 31). Similarly, MS-DIAL enabled to detect 9054 features and 9042 features had MS2 spectra extracted. This indicated both Met4DX and MS-DIAL enabled to process IM-AIF data from Agilent instrument. Altogether, we have demonstrated that Met4DX is a versatile data processing tool for various 4D metabolomics data with high-coverage and quantification precision in peak detection.

## Discussion

Ion mobility-resolved untargeted metabolomics provides the multidimensional analysis of the complex metabolome with improved resolving power and peak capacity, and generates four-dimensional data for accurate metabolite annotation[6,17]. However, the high dimensionality and complexity of four-dimensional data presents a great challenge to data processing, which results in low rates of peak detection and restricts its widespread applications. Currently, peak detection in 4D proteomics[18,19] and metabolomics[22] mostly converted 4D data into 3D data through the top-down-based dimensionality reduction. This strategy simplifies the data structure, but significantly reduced the peak detection sensitivity. In our study, we developed a fundamentally different strategy, which employs a MS spectrum-orientated bottom-up assembly algorithm for 4D peak detection in ion mobility-resolved untargeted metabolomics. Our method considers a MS spectrum as the smallest unit in 4D dataset, and builds its related elution peaks in IM dimension and LC dimension, respectively. Thus, the method processes 4D metabolomics data from the precursor spectrum to IM peak, LC peak, and integration of 4D peak in a reverse order of multidimensional separation. We refer to this as a reverse engineering strategy from data acquisition. With this algorithm, we developed Met4DX, an end-to-end computational tool for peak detection, quantification, and metabolite annotation in 4D metabolomics. With different biological samples, we demonstrated that Met4DX enabled high coverage and sensitivity of 4D peak detection, and achieved >90% of 4D peaks with high fidelity. We thoroughly benchmarked and validated Met4DX with other existing tools, including peak coverage, sensitivity and peak fidelity as well as quantification precision. We also demonstrated that Met4DX not only unleashed in-depth identification of metabolites but also achieved successful differentiation of co-eluted isomeric metabolites with high coverage and accuracy.

Isomeric metabolites commonly exist in metabolome and could not be fully resolved by conventional LC−MS. They tend to co-elute on chromatography, co-fragmentation and share similar fragments in MS/MS spectra, restricting the accurate metabolite identification in LC−MS-based untargeted metabolomics[3,4]. Ion mobility-resolved 4D metabolomics has demonstrated distinct advantages to separate metabolite isomers and generate four-dimensional data to characterize these isomers[17]. Although LC−IM−MS technology enables to separate these isomers, it is highly challenging for data processing software tools to detect and differentiate them from the complex 4D metabolomics data. In this work, we demonstrated that the bottom-up assembly based 4D peak detection in Met4DX had superiority and high sensitivity in detecting co-eluted isomeric metabolites (Fig. 5f−i and Supplementary Figs. 16−19). Small differences on both IM and LC dimensions were precisely recognized and differentiated. Due to this technological advancement, Met4DX increased the coverage of co-eluted isobaric features by 2−4 folds in real 4D metabolomics data of various biological samples. We also demonstrated that Met4DX enabled to discriminate co-eluted pairs of isobaric features with ΔCCS as small as 1%, which could only be partially separated by IM. These co-eluted isomeric metabolites are often failed to be detected by other software tools such as MS-DIAL and MetaboScape (Fig. 5f−i and Supplementary Fig. 21). Additionally, the detection of co-eluted isobaric feature pairs by Met4DX facilitated accurate annotation of metabolites, in particular for isomer metabolites. Notably, the resolving power of IM has been rapidly improved in the past decade. Some commercial and prototype IM-MS instruments (e.g., Cyclic IMS[31] and SLIMS[32]) achieved IM resolving power up to 500−1000. With these advancements, Met4DX will become more effective to discriminate isomeric metabolites in complex metabolome, which will empower the widespread applications of ion mobility-resolved 4D metabolomics.

Currently, MS/MS spectrum is crucial for metabolite annotation and structural elucidation in untargeted metabolomics. Deciphering MS/MS spectral data through tandem spectral match with standard library, network-based strategies (e.g., GNPS[33] and MetDNA[34]), and in silico prediction tools (e.g., SIRIUS[35] and MS-FINDER[36]) have readily facilitated metabolite annotation. With the development of Met4DX, it become feasible to expand more analytical dimensions to support multidimensional metabolite annotation including LC separation-derived RT and IM separation-derived CCS. The combination of LC and IM separations with MS measurements has been proved to improve resolving power, selectivity, and sensitivity, thereby benefiting the fidelity of metabolite annotation[6,17]. We demonstrated that Met4DX is particularly prominent to achieve high-coverage 4D peak detection with 4D information (MS1, RT, CCS, and MS/MS) of metabolites, maximizing information coverage to support in-depth metabolite annotation. We believe the integration of Met4DX with state-of-art metabolite annotation strategies, such as network-based tools (e.g., GNPS and MetDNA) and in silico prediction tools (e.g., SIRIUS) will further increase the coverage and accuracy of metabolite annotation in metabolomics. For 4D metabolomics, in the past decade, many CCS prediction tools[13,37,38] with high accuracy were developed (with CCS prediction errors of 1−2%). However, the CCS prediction accuracy highly depended on the coverage of training datasets and training models. For some compounds lacking similar structures in training datasets, these tools generated predicted CCS values with relatively large errors, introducing false positives/negatives during metabolite annotation. Alternatively, the curation of experimental CCS values from chemical standards is promising to achieve better performances in metabolite annotation. Compared with proteomics, error rate estimation (i.e., false discovery rate) is under development in metabolomics. Recently, DIAMetAlyzer[39] reported an automated false-discovery rate-controlled analysis for data-independent acquisition in metabolomics. More specifically for LC−IM−MS-based lipidomics, MS-DIAL reported the FDR of lipid annotations during RT and CCS

matches using their validation set[22]. Sterol4DAnalyzer developed by our group also reported the FDR of sterols under different CCS match tolerances[40]. A comprehensive FDR estimation for 4D metabolomics annotation should include a ground-truth benchmark dataset and evaluations of all 4-dimensional matches with different tolerances, which would be an important future plan for 4D metabolomics.

In summary, Met4DX is a mass spectrum-oriented end-to-end computational tool for 4D untargeted metabolomics, which deciphers the complex 4D information of metabolites in metabolomics data, and substantially advances the discovery of functional metabolites in biological organisms. Metabolomics software tools with broad utilization are usually developed with user-friendly graphical user interface (GUI) and support all-in-one solution from raw data importing to metabolite annotation, like MS-DIAL, MZmine[41], XCMS Online[42], and so on. A well-designed GUI enables users to adjust parameters easily and check the result with convenience. The current version of Met4DX could only be run as an R package using command lines. In the future, a GUI with comprehensive functions would make Met4DX more user-friendly.

## Methods

### Chemicals

LC−MS grade methanol (MeOH), 2-propanol (IPA), and water ($H_2O$) were purchased from Honeywell (Muskegon, MI, USA). LC−MS grade acetonitrile (ACN) was purchased from Merck (Darmstadt, Germany). Ammonium hydroxide ($NH_4OH$) and ammonium acetate ($NH_4OAc$) were purchased from Sigma-Aldrich (St. Louis, MO, USA). Metabolite chemical standards were purchased from Sigma-Aldrich (St. Louis, MO), J&K (Shanghai, China), and Bidepharm (Shanghai, China). The NIST human urine (SRM 3667) and NIST human plasma (SRM 1950) sample were purchased from Ango Biotechnology (Shanghai, China). Chemical standards for 20 natural products were purchased from TopScience (Shanghai, China).

### Sample preparation

The preparation of biological samples followed our published protocols[34,43]. For NIST human urine samples, 150 μL of urine was taken and 600 μL of MeOH was added for extraction. The mixture was vortexed for 30 s and sonicated for 10 min at 4 °C in water bath. To facilitate protein precipitation, samples were then incubated for 1 h at −20 °C, and centrifuged for 15 min at 17,000 × g and 4 °C. The supernatant was collected and evaporated to dryness at 4 °C. 150 μL of ACN/$H_2O$ (1:1, v/v) were added into the dry extracts for reconstitution. The solution was vortexed for 30 s, sonicated at 4 °C for 10 min, and centrifuged at 17,000 × g and 4 °C. Finally, the supernatant was transferred into a vial for LC−IM−MS analysis. To prepare serially diluted urine samples, the final supernatant was directly diluted with ACN/$H_2O$ (1:1, v/v) by 10, 100, and 1000 folds, respectively. For NIST human plasma samples, 150 μL of plasma was taken and 600 μL of MeOH/ACN (1:1, v/v) was added for extraction. The rest procedures were kept the same as described for urine samples. The 293T cell line was obtained from ATCC with Product No. CRL-2925. The cells were cultured in 6-cm dishes with Dulbecco modified Eagle's medium (DMEM) containing FBS (10%) and penicillin/streptomycin (1%). When growing to ~2,000,000 cells/dish, cells were harvested for metabolomics analysis. The culture medium was quickly removed and cells were washed with cold PBS twice. Dishes were placed on dry ice, and 1 mL of ACN/MeOH/$H_2O$ (2:2:1, v/v/v) was added for fast quenching and extraction. The extraction solution was pre-cooled at −80 °C for 1 h prior to the extraction. Then, dishes were incubated at −80 °C for 40 min. The cells were scraped from dishes and transferred to a 1.5-mL centrifuge tube. The samples were then vortexed for 1 min, and centrifugated for 15 min at 17,000 × g and 4 °C. The supernatant was collected and evaporated to dryness at 4 °C. For reconstitution, 100 μL of ACN/$H_2O$ (1:1, v/v) were added into the dry extracts. The rest of the procedure was kept the sample as described for NIST human urine samples. The

preparation of mouse liver and brain tissues followed our previous publication[34,43]. The natural products were dissolved in MeOH with a concentration of 1 mg/mL as the stock solution. To prepare the natural products spiked-in NIST human urine sample with the highest concentration (i.e., 1X dilution), we took 1 µL of each stock solution and spiked into 80 µL of reconstituted NIST human urine solution in ACN/$H_2O$ (1:1, v/v). Then, the 1X dilution sample was further diluted with reconstituted NIST human urine solution by 4, 16, 64, and 256 folds, respectively (Supplementary Fig. 11).

### LC−IM−MS analysis

All metabolomics data were acquired using an UHPLC system (1290 series, Agilent Technologies, USA) coupled to a timsTOF Pro equipped with an electrospray ionization (ESI) source (Bruker Daltonics, Bremen, Germany) or DTIM-MS 6560 (Agilent Technologies, USA). For hydrophilic interaction liquid chromatography (HILIC) separation, Waters ACQUITY UPLC BEH Amide column (particle size, 1.7 µm; 100 mm (length) × 2.1 mm (i.d.)) was used for the LC separation and the column temperature was kept at 25 °C. For both positive and negative modes, mobile phase A of 25 mM ammonium hydroxide ($NH_4OH$) and 25 mM ammonium acetate ($NH_4OAc$) in water and mobile phase B of ACN were used, respectively. The gradient was set as below: 0–0.5 min: 95% B, 0.5–7 min: 95% B to 65% B, 7–8 min: 65% B to 40% B, 8–9 min: 40% B, 9–9.1 min: 40% B to 95% B, and 9.1–12 min: 95% B. The flow rate was 0.5 mL/min. The injection volume was 3 µL. For reverse phase (RP) separation, Phenomenex Kinetex C18 column (particle size, 2.6 µm; 100 mm (length) × 2.1 mm (i.d.)) was used and the column temperature was kept at 25 °C. For both positive and negative modes, mobile phase A of 0.01% acetic acid (v/v) in water and mobile phase B of IPA/ACN (1:1, v/v) were used, respectively. The gradient was set as below: 0–1 min: 1% B, 1–8 min: 1% B to 99% B, 8–9 min: 99% B, 9–9.1 min: 99% B to 1% B, 9.1–12 min: 1% B. The flow rate was 0.3 mL/min. The injection volume was 3 µL. For MS acquisition, PASEF-DDA scan mode was applied with mass range from 20 to 1300 Da and mobility range from 0.45 to 1.45 V·s/cm². Detailed parameters for MS acquisition were set as follow: capillary voltages were set as +4500 V and −3600 V for positive and negative ionization modes, respectively; nebulizer pressure, 2.2 bar; dry gas, 10.0 L/min; dry temperature, 220 °C; number of PASEF MS/MS scans, 2; ramping time, 100 ms; TIMS stepping enabled; total cycle time, 0.53 s; charge range, 0–1; absolute threshold, 100 cts; active exclusion, checked; former target ions released after 0.1 min; isolation window, 1.2 Da; collision energy, 30 eV. In PASEF-DIA data acquisition, 7 mass steps with mass width of 144.3 Da and mass overlap of 5 Da were set in each cycle, covering the mass ranges of 20–1000 Da and mobility range of 0.45–1.45 V·s/cm². The cycle time is 0.64 s. The data acquisition was performed using timsControl (version 2.0, Bruker Daltonics, Bremen, Germany). For IM-AIF data acquisition in Agilent DTIM-MS, the source parameters were set as follows: sheath gas temperature, 325 °C; dry gas temperature, 300 °C; sheath gas flow, 11 L/min; dry gas flow, 8 L/min; capillary voltage, 4000 V; and nebulizer pressure, 20 psi. The TOF mass range was set as m/z 50–1700 Da. For ion mobility parameters, nitrogen ($N_2$) was used for the drift gas. Other related IM parameters were set as follows: entrance and exit voltages of drift tube, 1600 and 250 V; trap filling and trap release times, 20,000 and 150 µs. The pressure of drift tube was set at 3.95 Torr. The MS/MS spectra were acquired in the "Alternating frames" mode, and the collision energy was fixed at 20 V in frame 2. The CCS values were calculated with single electric field method. Data acquisitions were carried out using MassHunter Workstation Data Acquisition Software (Version B.08.00, Agilent Technologies, USA).

### Curation of 4D metabolite library

All metabolites in KEGG (accessed on 7 March, 2017; https://www.genome.jp/kegg/) and HMDB (accessed on 8 November, 2021; https://hmdb.ca/) were combined and dereplicated with their InChIKey. Then, metabolites with exact masses larger than 1200 Da or less than 60 Da were removed. Compounds without available SMILES were removed. Finally, a total of 135,638 unique metabolites were generated as the Met4DX metabolite library (Supplementary Fig. 12a). The 4D information for each metabolite was generated as follows and organized in a msp format file.

**MS1 dimension.** With the formula and adduct information, we calculated theoretical m/z of metabolite ions, including $[M + H]^+$, $[M + Na]^+$, $[M + NH_4]^+$ and $[M + H\text{-}H_2O]^+$ for positive mode, and $[M\text{-}H]^-$, $[M + Na\text{-}2H]^-$, $[M + HCOO]^-$ for negative mode.

**CCS dimension.** The SMILES strings of metabolites were inputted in our previous developed AllCCS (http://allccs.zhulab.cn/) for calculation CCS values in different adduct forms[13]. Specifically, $[M + H]^+$, $[M + Na]^+$, $[M + NH_4]^+$ and $[M + H\text{-}H_2O]^+$ for positive mode, and $[M\text{-}H]^-$, $[M + Na\text{-}2H]^-$, $[M + HCOO]^-$ for negative mode were calculated, respectively.

**RT dimension.** We curated the RT library through experimental measurements and in silico prediction. We experimentally measured 883 and 907 RT values on hydrophilic interaction liquid chromatography (HILIC) and reverse phase (RP) separations, respectively, covering a total of 1014 metabolites. For other metabolites, we employed a graph neural network (GNN)-based RT prediction model developed by Lu group for RT prediction[44,45]. The GNN-RT models on RP and HILIC separations were trained with the SMRT dataset from Siuzdak group[46] and the Retip dataset from Fiehn group[47], respectively. In Met4DX, we performed the transfer learning of RT predictions using the experimental measured HILIC and RP RTs in our LC conditions, respectively, and constructed two new GNN-RT prediction models. In transfer learning, key parameters such as the frozen layers number, learning rate as well as the iteration number were optimized during the training of transfer learning. In the testing of transfer learning, the median absolute errors were 17.62 and 9.65 s in HILIC and RP separations, respectively (Supplementary Fig. 32). Finally, the GNN-RT prediction models were used to predict RTs for metabolites without experimental RTs.

**MS/MS dimension.** We first acquired the standard MS/MS spectra for 1064 metabolites with the same IM-MS parameters described in LC−IM−MS analysis part. In addition, we collected the experimental MS2 spectra in MoNA (https://mona.fiehnlab.ucdavis.edu/; accessed on 31 May, 2022) and NIST 20 mass spectral library, and dereplicated the spectra using InChIKey. A total of 5234 and 2784 metabolites had experimental MS2 spectra in positive and negative modes, respectively, covering 5506 metabolites. To predict MS2 spectra, in Met4DX, the candidate metabolites after MS1, RT, and CCS matches were further inputted into MS-FINDER (version 3.24) for prediction of MS2 spectra and MS2 spectral match (scoring and ranking).

### The workflow of Met4DX

The Met4DX workflow includes four major modules: (1) MS2 spectral dereplication; (2) the bottom-up assembly algorithm for 4D peak detection; (3) 4D peak alignment and grouping; (4) multidimensional match for metabolite identification.

(1) *MS2 spectral dereplication*

    For each data file, MS2 spectral dereplication was performed. Each MS2 spectrum was first purified by removing ions with intensities less than 30 and lower than 1%. Then, MS2 spectra were binned according to their differences of precursor m/z, RT, and mobility (m/z tolerance, 20 ppm or 0.004 Da for m/z < 200 Da; RT tolerance, 20 s; mobility tolerance, 0.030 V·s/cm²). For bins with more than 1 MS2 spectrum, 3D distances of

all pairwise MS2 spectra were calculated by integrating the distances of precursor RT, mobility, and MS2 spectral similarity with Eqs. (1–6). The trapezoidal function was used to calculate the distances of RT and mobility, while the dot-product function was used to calculate the distance of MS2 spectra. A hierarchical cluster analysis (HCA) was applied to cluster MS2 spectra in the bin. Here, the cutoff of 3D distance was set as 1. For each MS2 cluster, the most intense MS2 spectrum (i.e., the unique MS2 spectrum) was selected and outputted to represent the cluster. The spectral intensity in an MS2 spectrum is the sum intensity of top 10 fragment ions ranked by their intensities.

$$\text{Dist}_{i,j} = \begin{cases} 0, & \Delta RT \text{ or } \Delta \text{mobility} \leq TOL_{min} \\ \frac{(\Delta - TOL_{min})}{TOL_{max} - TOL_{min}}, & TOL_{min} < \Delta RT \text{ or } \Delta \text{mobility} \leq TOL_{max} \\ 1, & \Delta RT \text{ or } \Delta \text{mobility} > TOL_{max} \end{cases} \quad (1)$$

$$\Delta RT_{i,j} = |RT_i - RT_j| \quad (2)$$

$$\Delta \text{mobility}_{i,j} = |\text{mobility}_i - \text{mobility}_j| \quad (3)$$

$$\text{Dist}_{MS2_{i,j}} = 1 - \frac{\Sigma W_i W_j}{\sqrt{\Sigma W_i^2 W_j^2}} \quad (4)$$

$$W = \text{Intensity}_{fragment}^1 \times [m/z]_{fragment}^0 \quad (5)$$

$$\text{Dist}_{3D_{i,j}} = \sqrt[2]{W_{RT} \times \text{Dist}_{RT_{i,j}}^2 + W_{mobility} \times \text{Dist}_{mobility_{i,j}}^2 + W_{MS2} \times \text{Dist}_{MS2_{i,j}}^2} \quad (6)$$

where $i$ and $j$ were the indexes of MS2 spectra in the bin and $i \neq j$. The maximum tolerance ($TOL_{max}$) was the acceptable tolerance, while the minimum tolerance ($TOL_{min}$) was the penalty-free tolerance. Here, $TOL_{max}$ and $TOL_{min}$ were set as 10 s and 20 s for RT, and 0.015 and 0.030 V·s/cm$^2$ for mobility, respectively. W was the weight in each dimension, and all weights were set as 1.

(2) *The bottom-up assembly algorithm for 4D peak detection*

For each MS data file, Met4DX assembled the 4D peaks from its unique MS2 spectra with a bottom-up assembly strategy. To access raw data in.d format, the R package opentimsr[48] (https://github.com/cran/opentimsr; version 1.0.13) and Bruker TDF-SDK (https://www.bruker.com/protected/zh/services/software-downloads/mass-spectrometry/raw-data-access-libraries.html; version 2.8.7.1) were used. After querying the raw data and extracting all MS1 frames, a 5-step 4D peak detection was implemented as follows:

Step 1: precursor search. For each unique MS2 spectrum, Met4DX searched its precursor MS1 data point in the $m/z$-ion mobility data frame with the precursor $m/z$, ion mobility, and frame index recorded in the .mgf file. Step 2: EIM detection. In the precursor frame, the adjacent MS1 data points within the set $m/z$ and mobility ranges were retrieved to reconstruct the ion mobilogram. Specifically, $m/z$ tolerance was 20 ppm or 0.004 Da for $m/z$ < 200 Da, and the range to reconstruct ion mobilogram was set as 0.1 V·s/cm$^2$ around the precursor MS1 data point. Then, the locally weighted scatterplot smoothing (LOESS) was applied for data smoothing. The EIM peak was detected by finding the local maximum within the peak span. The default peak span of EIM detection was 13 data points, converting to 0.013 V·s/cm$^2$ under our instrument parameters. If multiple apexes found, the EIM peak with a smallest mobility difference to the precursor mobility was selected. Step 3: EIM extension. If EIM detection in step 2 was successfully achieved

in the precursor frame, this step further extended EIM detection in 30 adjacent MS1 frames around precursor frames. Step 4: EIC detection. The EIM peaks in each frame were summed up to generate the frame intensities, which were further projected to the LC dimension. Thus, the ion chromatogram on LC dimension was assembled. After LOESS smoothing, the EIC peak was also detected by finding the local maximum within the peak span. The default peak span of EIC detection was 11 data points, converting to ~5.3 s in our experimental conditions. If multiple apexes found, the EIC with a smallest RT difference to the precursor RT was selected. Step 5: 4D integration. With the EIM peaks detected in step 2 and the EIC peak detected in step 4, the 4D peak was constructed. Met4DX integrated the ion intensities around the apexes. Specifically, the MS1 data points in 5 frames nearest to the EIC apex and within mobility differences of 0.015 V·s/cm$^2$ to the EIM apexes were integrated. The $m/z$ value of the 4D peak was re-weighted with the integrated MS1 data points' $m/z$ values and intensities.

In 4D peak detection, a series of criteria were implemented to ensure the peak fidelity. During the EIM and EIC peak detection, the apexes should be the local maximum within the set span in accordance with the peak width at half height under our LC–IM–MS parameters. The detected apex should be within the tolerances of the precursor RT or mobility, respectively. Specifically, 10 s and 0.015 V·s/cm$^2$ were used, respectively. Additionally, there should be continuous signals near the apex. The signal-to-noise ratios (S/N) were also calculated to filter those signals with S/N lower than 3. Furthermore, noisy signals with high fluctuation (normalized standard noise ≥0.35) were also filtered, which followed our previous publication[49].

After the completion of the 4D peak detection, a 4D peak table was generated for each MS file. In this table, Met4DX calculated the CCS value of each feature from its experimental mobility with Eq. (7):

$$CCS = \text{convertor}^* \frac{z}{K_0} \sqrt{\frac{1}{TM}} \sqrt{\frac{M+m}{m}} \quad (7)$$

where $z$ was the ion charge and $1/K_0$ was the mobility. $T$ is the temperature, 305 K. $M$ and $m$ are exact masses of the analyst ion and N$_2$, respectively[50].

(3) *4D peak alignment and grouping*

We used a landmark-based strategy for RT alignment. The detected 4D peaks between one specific reference sample and other samples were matched with 4D information ($m/z$, RT, CCS, and MS2 spectra). The reference sample was user-defined. We recommended using one of the pooled quality control (QC) samples or the middle sample in the injection order as the reference sample. Then, a RT correction model was built with RTs of landmarks using a LOESS-based strategy. The RTs in samples were corrected and aligned to the reference sample. After that, a density-based peak grouping strategy[51] was implemented for feature grouping. 4D features across samples were binned together according to their $m/z$ and mobility values. The bin sizes were set as 0.015 Da and 0.015 V·s/cm$^2$. The density on RTs was profiled to generate the final 4D peak group. The bandwidth of gaussian smoothing was set as 5 s. The MS2 spectral stochasticity is common in DDA-based data acquisition. Met4DX re-performed the 4D peak detection in samples with missing peaks. The medium values of $m/z$ and mobility among peaks in the peak group were used as the precursor information to perform the targeted 4D peak detection in samples with missing peaks. The MS1 frame with the smallest RT difference to the median of RTs in the peak groups was regarded as the precursor frame. The re-assembled 4D peaks should also meet the criteria

of peak fidelity. Peak groups those met the requirement of minimal fraction (min. frac.; 0.5 by default) were kept. Gap filling was applied for missing values in the peak table through the mandatory integration, which integrated the signals in 5 frames, 20 ppm, and 0.015 V·s/cm$^2$ in mobility near the RT, $m/z$, and mobility values of the 4D peak. Met4DX finally outputs the 4D peak table with the qualitative information of $m/z$, RT, and CCS as well as peak intensities in a.csv table, named "features_filled.csv". MS2 spectra for features were also exported in a .msp file, named "spectra.msp".

(4)  *Multidimensional match for metabolite identification*

Met4DX performed multidimensional match between each feature and the 4D metabolite library. First, Met4DX searched candidate metabolites in the library by sequentially matching MS1, RT, and CCS values. The $m/z$ tolerance was set as 20 ppm in MS1 match. RTs in the metabolite library were first recalibrated into the experimental condition according to the protocol in our previous publication[43,52], then matched to experimental values of 4D features. Finally, CCS match was performed. In RT and CCS matches, the minimum and maximum tolerances of trapezoidal function were set as 30 s and 90 s in RT match, and 3% and 6% in CCS match, respectively (Eqs. 8–10). The CCS match tolerances were set according to the CCS prediction accuracy of AllCCS[13]. The results showed that 72% of predicted CCS values had relative error less than 3% while 93% of predicted CCS values had relative error less than 6% (Supplementary Fig. 33). The CCS match tolerances were user-defined. If the user inputs a library with CCS values with high accuracy, the tolerances could be reduced. Only metabolite candidates within the maximum tolerance were remained and scored.

$$\text{Score} = \begin{cases} 1, & \Delta\text{RT or }\Delta\text{CCS} \le \text{TOL}_{\min} \\ 1 - \frac{(\Delta - \text{TOL}_{\min})}{\text{TOL}_{\max} - \text{TOL}_{\min}}, & \text{TOL}_{\min} < \Delta\text{RT or }\Delta\text{CCS} \le \text{TOL}_{\max} \\ 0, & \Delta\text{RT or }\Delta\text{CCS} > \text{TOL}_{\max} \end{cases} \quad (8)$$

$$\Delta\text{RT} = |\text{RT}_{\text{experiment}} - \text{RT}_{\text{library}}| \quad (9)$$

$$\Delta\text{CCS} = \frac{|\text{CCS}_{\text{experiment}} - \text{CCS}_{\text{library}}|}{\text{CCS}_{\text{library}}} \times 100\% \quad (10)$$

For candidate metabolites having experimental MS2 spectra in library, MS2 spectral match was scored with the dot-product function (Eqs. (11) and (5)). The cutoff value was set as 0.8. For matched candidate metabolites, the match scores in RT match, CCS match, and MS2 spectral match were integrated to calculate the final combined score using a linear weighting function (Eq. (12)). The weighs of RT match, CCS match, and MS2 spectral match were set as 0.2, 0.4, and 0.4, respectively. The cutoff value of the combined score was set as 0.6.

$$\text{Score}_{\text{MS2}} = \frac{\Sigma W_{\text{experiment}} W_{\text{library}}}{\sqrt{\Sigma W_{\text{experiment}}^2 W_{\text{library}}^2}} \quad (11)$$

$$\text{Score} = W_{\text{RT}} \times \text{Score}_{\text{RT}} + W_{\text{CCS}} \times \text{Score}_{\text{CCS}} + W_{\text{MS2}} \times \text{Score}_{\text{MS2}} \quad (12)$$

For candidate metabolites without experimental MS2 spectra in the metabolite library, Met4DX exported the feature and its experimental MS2 spectrum into MS-FINDER for structural scoring and ranking. In Met4DX, it only kept the top 3 candidates in [M + H]$^+$ or [M-H]$^-$ adduct forms outputted by MS-FINDER. Finally, all metabolite annotation results outputted in a table named "ScoreCombine.csv".

## Data processing with Met4DX

For PASEF-DDA data, MS2 spectra in each raw MS file were first converted into the .mgf files using DataAnalysis (Bruker Daltonics, Bremen, Germany, version 5.2). The raw MS data files and MS2 spectra in mgf files were organized into one folder and imported into Met4DX for data processing. If MS2 spectral files were not provided, Met4DX enabled to generate MS2 spectra directly from the raw data files using the "GenerateMS2" function. Our data showed that Met4DX showed high consistency in peak detection using MS2 spectra converted from DataAnalysis and generated by the "GenerateMS2" function (Supplementary Fig. 34). The demo code to use the "GenerateMS2" function was provided in GitHub (https://github.com/ZhuMetLab/Met4DX). Detailed parameters were provided in Supplementary Table 1. Met4DX finally outputted the 4D peak table with the qualitative information of $m/z$, RT, and CCS as well as peak intensities in a .csv table, named as "features_filled.csv". MS2 spectra for 4D features were exported in a .msp file, named as "spectra.msp". If metabolite annotation was performed, the multidimensional match result was outputted as a "ScoreCombine.csv" file. Met4DX also supported metabolite annotation with a user-defined metabolite library, and the instruction was provided in Supplementary Note 1.

For PASEF-DDA data, Met4DX also supports the input of a precursor ion list as seeds to initiate 4D peak detection. In Met4DX, we have curated a list of precursor ions collected from various biological samples ($N = 72{,}265$ in positive mode; $N = 42{,}553$ in negative mode; Supplementary Data 9) for this workflow. Each ion included $m/z$, RT, and CCS information. First, Met4DX converted CCS values in the ion list into mobility values. For each of the inputted precursor ions, the bottom-up assembly algorithm was performed for 4D peak detection (see the workflow of Met4DX for details). Then, Met4DX generated a 4D peak table after peak grouping which was further filtered with a minimal fraction (min. frac.) of 0.5. To assign MS2 spectra for each feature, Met4DX matched the precursor ion information ($m/z$, rt, and mobility) of an MS2 spectrum to those of 4D features, and selected the most intense MS2 spectrum. Finally, the gap-filling was performed in samples with missing values. Detailed parameters were provided in Supplementary Table 2.

For PASEF-DIA data, Met4DX also employs the user inputted precursor ion list to initiate 4D peak detection. The detailed workflow is similar to the processing of PASEF-DDA data. The major difference is the assignment of MS2 spectra. For each 4D peak, Met4DX selected the sample with the highest peak intensity. Then, the MS2 frame at the retention time apex of the 4D peak was retrieved in the specific sample. In this $m/z$-mobility frame for MS2 spectra, fragment ions at mobility apex were extracted and evaluated whether they were true signals with continuous data points along mobility axis and similar $m/z$ values. Finally, an MS2 spectrum was generated at the apex of retention time and mobility and assigned to the 4D peak. Detailed parameters were provided in Supplementary Table 3.

For Agilent IM-AIF data, raw data files (.d format) were first converted into .mzML format with ProteoWizard (version 3.0.20360). Then, Met4DX converted CCS values in the precursor ion list into drift times with CCS calibration coefficients ($t_{\text{fix}}$ and β). For each precursor ion, the bottom-up assembly algorithm was performed for 4D peak detection (see the workflow of Met4DX for details). Met4DX generated a 4D peak table after peak grouping, which was further filtered with a minimal fraction (min. frac.) of 0.5. The extraction of MS2 spectra at apex of retention time and drift time was also performed to assign MS2 spectra to each 4D peak. Finally, the gap-filling was performed in samples with missing values. Detailed parameters were provided in Supplementary Table 4.

## Data processing using MS-DIAL and MetaboScape

For MS-DIAL (version 4.60 for Bruker PASEF-DDA and version 4.90 for Bruker PASEF-DIA and Agilent IM-AIF data processing), raw data files

were first converted into .ibf format with IbfConverter.exe. Then, MS-DIAL performed peak detection, alignment, gap filling, and MS2 spectral assignment with the converted data files. Detailed parameters were provided in Supplementary Table 5. The feature table with representative MS2 spectra was outputted as a .txt file. After removing features with −1 in the "Average mobility" column, the rest features were counted as the 4D features. The 4D features with MS2 spectra in the "MS/MS spectrum" column were regarded as the complete 4D features in MS-DIAL. These 4D features were used for benchmark in our study. For MetaboScape (for PASEF-DDA and PASEF-DIA data processing; Bruker Daltonics, Bremen, Germany; version 2022b), raw data files were imported for peak detection, alignment, and MS2 spectral assignment. Detailed parameters were provided in Supplementary Table 6. All 4D features were selected and exported to a .csv table. 4D features with TRUE in the "MS/MS" column were regarded as complete 4D features assigned with MS2 spectra. These 4D features were used for benchmark in our study. For feature overlapping comparison in different software tools, $m/z$ tolerance, RT, and CCS tolerances were set as 20 ppm, 10 s, and 2%, respectively.

### Evaluation of peak fidelity using EVA
For each 4D feature with MS2 spectra from Met4DX, MS-DIAL, and MetaboScape, we selected the replicate sample with the highest intensity, and then assembled the 4D peak in this replicate sample to obtain the EIC and EIM peaks and evaluate the peak fidelity of the feature. For EIC peaks, the IM dimension was removed, while for EIM peaks, the LC dimension was removed. The EIC and EIM peak shapes were inputted into the EVA software for peak fidelity evaluation[26]. Specifically, in MS-DIAL and MetaboScape, we used the feature information from their feature tables to re-extract the 4D features for peak fidelity evaluation. The peak fidelity rates were calculated with these re-extracted 4D features (Supplementary Fig. 8).

### Reporting summary
Further information on research design is available in the Nature Portfolio Reporting Summary linked to this article.

## Data availability
All raw data files of biological samples acquired by TIMS with PASEF-DDA can be assessed by National Omics Data Encyclopedia under Project ID OEP003701 and Zenodo [https://doi.org/10.5281/zenodo.7215544][53]. Raw data files of NIST human urine sample acquired by TIMS with PASEF-DIA and DTIM-MS with IM-AIF can be assessed by National Omics Data Encyclopedia under Project ID: OEP003846. KEGG database (https://www.genome.jp/kegg/) was accessed on 7 March, 2017. HMDB database (https://hmdb.ca/) was accessed on 8 November, 2021. Source data are provided with this paper.

## Code availability
The source code of Met4DX was provided in GitHub (https://github.com/ZhuMetLab/Met4DX) and Zenodo (https://doi.org/10.5281/zenodo.7701165)[54].

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

## Acknowledgements

The work was supported by National Natural Science Foundation of China (31971356 to Z.J.Z.), National Key R&D Program of China (2018YFA0800902 to Z.J.Z.), Excellent Young Scholar Fund from NSFC (22022411 to Z.J.Z.), Shanghai Municipal Science and Technology Major Project (2019SHZDZX02 to Z.J.Z.), and Shanghai Key Laboratory of Aging Studies (19DZ2260400 to Z.J.Z.).

## Author contributions

Z.J.Z., M.L., and Y.Y. conceived the idea and designed the algorithm and software. Y.Y., M.L., and X.C. tested and debugged the package. M.L. performed the sample preparation, data acquisition, data processing, and data analysis. Z.Z. developed the multidimensional match algorithm and contributed to the data analysis. H.Z. curated the experimental RT and MS/MS spectral libraries of metabolites. H.W. contributed to the RT prediction. Z.J.Z. and M.L. wrote the manuscript. Z.J.Z. supervised the project.

## Competing interests

The authors declare no competing interests.
