## [Peer Review File · Nature Communications]

REVIEWER COMMENTS

Reviewer #1 (Remarks to the Author):

This work describes a peak detection and alignment approach for 4D LC-IMS-MS data. This is a much needed development, and I am excited for the further development of tools such as this. While there are many strengths to this tool, there are also some weaknesses and questions which have arisen as I read this paper. Hopefully my comments will help to improve not only the paper, but also the program, and in doing so metabolomics more generally. This is a great development which would be even greater were it more versatile.

It appears the 4D peak detection needs to be seeded by a precursor m/z - rt - IMS value. Can peak detection be performed in the absence of a precursor seed value? Alternatively, what if the user supplied a list of mz/rt values with or without IMS data? As it stands it appears that this program works only on Bruker files, (based on GitHub documentation and the premise of the program), but it could certainly be made to work with other vendor data if the seed step was more versatile than a PASEF DDA MS/MS precursor ion. The other advantage to diversifying is that you are currently limiting yourself ONLY to those precursors that have been sampled by DDA, which is unlikely to actually be a comprehensive sampling, even with PASEF. What happens if you move to PASEF DIA? It would seem that Met4Dx is incompatible with this data acquisition approach? If you are building a tool for one instrument/acquisition approach it will necessarily have a limited scope of application.

Figure 2e - what do the open circle/closed circle represent?

2-3 fold more peaks than MS-Dial or Metaboscape. How do we know these are real signals? Would adjusting the sensitivity settings in MS-Dial (for example, the 300 minimum amplitude setting) enable higher feature counts? Does the fidelity drop for those peaks that Met4Dx detects but are missed by MS-Dial, for example? What does 'manually checked' mean. Looking at 7000+ manual 4D integration peaks per injection? I doubt that can be done well, and it is probably not what you actually did.

Figure 5b - what do the three bars represent? No legend for colors

Figure 5c - what does the $N = 16M$ refer to? Are these isomeric feature pairs? Why would you scale the IMS CCS values (4, 2, 0.5%) but not the retention time values? Clearly with better chromatographic resolution you would also improve your identification statistics. Further, IMS has poor orthogonality to m/z relative to reverse phase LC.

Figure 5d - it would be useful to show some examples where MS/Dial/Metaboscape failed to pick these up. (maybe this is Supplementary Figure 17)??

Figure 5e - what does it look like if you do the same for retention time? ie. ppm < 10 ppm, but IMS values are the same with different retention times?

There is a fair amount of language claiming that IMS is great for isomer separation in a manner that chromatography isn't. I do not see data supporting this perspective in this paper, and published literature suggests that IMS is less orthogonal to mass than retention time. IMS does show a great deal of potential, but the presentation seems unrealistically optimistic to me. Increasing your chromatographic resolution also improves isomer separation! IMS is faster than chromatography but also occurs postionization, meaning that chromatography is necessary to reduce ionization suppression. Overselling IMS doesn't help. For example, figure 17h demonstrates an example 1% ims separation - most analytical chemists would not consider these to be resolved. Maybe a 2D blot (rt and ims) would help visualize this? 17g represents a 2-3% separation - at first glance it appears the chromatography may actually offer > 2-3% separation - why highlight IMS here?

Computational time?

EVA appears from the publication to be trained to estimate quality for 3D (LC-MS-Intensity) peaks. Does it work for 4D data, or did you need to remove a dimension to enable this evaluation?

Reviewer #2 (Remarks to the Author):

The manuscript with title "A mass spectrum-oriented computational method for ion mobility-resolved untargeted metabolomics" presents Met4DX, an end-to-end computational framework which performs peak detection, quantification, and identification of metabolites in IM-resolved 4D metabolomics with data-dependent acquisition. The peak detection strategy represents a great alternative to existing tools. I fully agree that more informatic solutions are needed to promote widespread application of 4D untargeted metabolomics. The software was implemented in R, the source code was provided with an accompanying container and instructions to use it. The performance of Met4DX was benchmarked against other existing tools using different standard complex biological samples. The manuscript is overall well written, and the results well presented. The following points should be addressed to improve the clarity of the work:

1. The Abstract and Introduction should specify that the workflow is for data-dependent acquisition: IM-resolved 4D data-dependent acquisition metabolomics.

2. Met4DX Workflow.

a. How is defined the intensity in the MS2 spectrum with the highest spectral intensity to be selected to represent the cluster? The sum of all MS2 peaks?

b. Limitation, Met4DX requires MS2, how many features could be detected by MS-DIAL and missed by Met4DX? Discuss this limitation of the Met4DX workflow. While the Met4DX approach provides advantages, many metabolites could be detected without any MS2 (either due to poor fragmentation of the specific molecule or to low concentration), but still the annotation based on m/z, RT and CCS could provide biological insights.

3. Figure 1-b. In the first panel, what do the equal sign and the dot mean? Add next to it the text "Precursor m/z, RT, CCS" for clarity.

4. Performance benchmark.

a. MetaboScape. Indicate the first time it is mentioned and in the Methods that this is a proprietary software from Bruker.

b. Can you add a few sentences about benchmarking performance regarding computing resources (run time and memory used)?

5. It seems the tool used to generate the required MGF files, DataAnalysis, is also a Bruker software, this could potentially hinder the utilization. If it is a proprietary software, can you provide an alternative using free/open-source software? Clarify this point in the manuscript.

6. Quantification precision. Perform an evaluation of the quantitation linearity using the dilution data (serially diluted urine samples). This will better demonstrate the accuracy of the peak integration in Met4DX, beyond number of IDs and comparison by pair of replicates. For example, see DIAMetAlyzer 10.1038/s41467-022-29006-z, Figure 3-c.

7. The methods indicate that a library containing predicted values of CCS, RT and MS/MS was generated for annotation. This could be a valuable resource for the community. Is it included in the container? If not, please share the library and specify in the manuscript and instructions how to use it. Furthermore, specify how the users can utilize their own library in Met4DX.

8. Figure 5-b: indicate in a better way the confidence levels. For example, if the levels are the colors related to figure 5-a, add a rectangle joining a and b, or indicate it in the figure legend. Furthermore, consider including Supplementary Figure 10 as part of figure 5.

9. Figure 5-d. What was the CCS tolerance to evaluate the number of co-eluted isobaric pair features detected by the different software tools?

10. Related to CCS tolerance, the Methods indicate that the minimum and maximum tolerances of trapezoidal function for CCS match were set as 3% and 6%, but Figure 5-c compares Δ CCS set as $\geq 4\%$, $\geq 2\%$, and $\geq 0.5\%$. Clarify these differences and how ambiguity is resolved when different annotations are matched to the same feature, or the same metabolite is matched to different features. Also, discuss the limitation of using predicted CCS values which have larger errors than the experimental values that can be obtained in current IM instrumentation.

11. Related to limitations, discuss the importance and future plans to include an error rate estimation method (i.e., false discovery rate) and a graphical-user interface (user-friendly tools are paramount to truly promote widespread application of 4D untargeted metabolomics).

12. What are the columns of the multidimensional match results in the "ScoreCombine.csv" file? Do you output only the combined score from the linear weighting function? Please also output each individual score. Do you output the confidence level indicating the type of reference for each dimension (i.e., experimental or predicted)?

Minor:

1. Check missing articles throughout the text. For example, in the Introduction, add "the": compositional diversity of "the" metabolome.

2. Discussion. Remove "of" in "and generates of four-dimensional data". Rephrase the last part in "the coverage and accuracy of annotated metabolites in untargeted metabolomics have remained to be improved".

Response to the reviewers:

The authors would like to thank the reviewers for the helpful comments. We feel these comments have strengthened the manuscript considerably.

Reviewer #1:

“This work describes a peak detection and alignment approach for 4D LC-IMS-MS data. This is a much needed development, and i am excited for the further development of tools such as this. While there are many strengths to this tool, there are also some weaknesses and questions which have arisen as i read this paper. Hopefully my comments will help to improve not only the paper, but also the program, and in doing so metabolomics more generally. This is a great development which would be even greater were it more versatile.”

Ans: We appreciate the reviewer’s positive comments towards publication. We have thoroughly revised the manuscript according to the comments. We believe that these comments have significantly improved the manuscript and our Met4DX software tool.

Comment #1: *“it appears the 4D peak detection needs to be seeded by a precursor m/z - rt - IMS value. Can peak detection be performed in the absense of a precursor seed value? Alternatively, what if the user supplied a list of mz/rt values with or without IMS data? As it stands it appears that this program works only on bruker files, (based on github documentation and the premise of the program), but it could certainly be made to work with other vendor data if the seed step was more versatile that a PASEF DDA MS/MS precursor ion. The other advantange to diversifying is that you are currently limiting your self ONLY to those precursors that have been sampled by DDA, which is unlikely to actually be a comprehensive sampling, even with PASEF. What happens if you move to PASEF DIA? It would seem that Met4Dx is incompatible with this data acquisition approach? If you are building a tool for one instrument/acquisition approach it will necessarily have a limited scope of application.”*

Ans: Thanks a lot for the reviewer’s comment. We agreed with the reviewer that the original 4D peak detection in Met4DX required a precursor ion list as “seeds” to initiate he mass spectrum-oriented bottom-up assembly algorithm. Originally, the precursor ion list was retrieved from MS2 spectra. This setting restricted the 4D peak detection only applicable for PASEF-DDA data from Bruker TIMS instrument.

In the revised manuscript, we followed the reviewer’s great suggestion, and modified Met4DX to support the input of **a list of precursor ions** as “seeds” to initiate 4D peak detection (**Supplementary Figure 26 and 27**). In revised version of Met4DX, we have curated a list of precursor ions collected from various biological samples (N=72,265 in positive mode; N= 42,553 in negative mode; **Supplementary Data 9**). Each ion includes *m/z*, RT, and CCS information. With the inputted precursor ion list, Met4DX enables to perform 4D peak detection and data processing for different IM-MS data including **PASEF-DDA and PASEF-DIA data from Bruker TIMS, and IM-AIF data from Agilent DTIM-MS (Supplementary Figure 28, 30 and31)**. Please read details in the following section.

Supplementary Figure 26. Met4DX supports peak detection using precursor ions retrieved from MS2 spectra and a user inputted ions of interest list.

Supplementary Figure 27. The Met4DX workflow using a user-inputted list of precursor ions.

First, to demonstrate the peak detection performances of Met4DX using the inputted list of precursor ions, we took human urine samples as examples, and compared the coverage and quantification precision with MetaboScape and MS-DIAL (**Supplementary Figure 28**). Met4DX detected a total of 11419 features in total. Among them, 8309 features (73%) had MS2 spectra (**Supplementary Figure 28a**). As a comparison, MS-DIAL and MetaboScape detected 14440 and 4351 features, and 3741 (26%) and 3927 (90%) features had MS2 spectra, respectively. The percentage of 4D features with MS2 spectra in MS-DIAL was significantly lower than Met4DX and MetaboScape. Moreover, Met4DX also showed higher quantification precision with a median RSD of 16.5%. The percentage of detected peaks with RSD less 30% was as high as 75% in Met4DX (**Supplementary Figure 28b**). As a sharp contrast, although MS-DIAL generated more 4D features, most of them had poor quantification precision (median RSD=47.8%; **Supplementary Figure 28c**).

Supplementary Figure 28. High-coverage and high quantification precision of 4D peak detection in Met4DX using the inputted precursor ion list. The data was acquired from NIST human urine samples with PASEF-DDA in positive mode (n=6 technical replicates).

Supplementary Figure 30. Met4DX enabled high-coverage 4D peak detection and MS2 spectral extraction in PASEF-DIA data compared with other software tools. The data was acquired from NIST human urine samples with PASEF-DIA in positive mode (n=6 technical replicates).

Second, we demonstrated the capability of Met4DX to process **PASEF-DIA** data from Bruker TIMS (**Supplementary Figure 30**) and **IM-AIF** data from Agilent DTIM-MS instruments (**Supplementary Figure 31**). In NIST human urine data acquired by PASEF-DIA from TIMS, Met4DX detected 10021 features. Among them, 7882 features (79%) had MS2 spectra extracted (**Supplementary Figure 30a**). As a comparison, for the same data set, MS-DIAL detected 10681 features, but only 104 features had MS2 spectra extracted (<1%, **Supplementary Figure 30b**). MetaboScape detected 5753 features and did not support MS2 spectral extraction for DIA data (**Supplementary Figure 30c**). Together, these results demonstrated that Me4DX enabled high-coverage 4D peak detection and MS2 spectral extraction in PASEF-DIA data compared with other software tools. Both MS-DIAL and MetaboScape have to improve MS2 spectral extraction function for PASEF-DIA data. In addition, for NIST human urine data acquired from Agilent DTIM-MS (with IM-AIF mode), Met4DX detected 8324 features, and 8318 features had MS2 spectra extracted (**Supplementary Figure 31**). Similarly, MS-DIAL enabled to detect 9054 features and 9042 features had MS2 spectra extracted. This indicated both Met4DX and MS-DIAL enabled to process IM-AIF data from Agilent instrument.

Supplementary Figure 31. The processing of IM-AIF metabolomics data acquired using Agilent DTIM-MS instrument using Met4DX (a) and MS-DIAL (b). The data was acquired from NIST human urine samples in positive mode (n=6 technical replicates).

Taken altogether, we have added a new paragraph in the revised manuscript to describe the related results:

“Versatile data processing in Met4DX. Met4DX is a versatile tool to process various types of IM-based 4D metabolomics data. In addition to process PASEF-DDA data starting from MS2 spectra, Met4DX also enables to process data with a user-inputted list of precursor ions to initiate the bottom-up assembly 4D peak detection (**Supplementary Figure 26 and 27**), which further increases the coverage of peak detection, in particular for those without MS2 spectra acquired. To facilitate this workflow, we have curated a list of precursor ions collected from various biological samples reported above (N=72,265 in positive mode; N= 42,553 in negative mode; **Supplementary Data 9**). Each ion includes m/z, RT, and CCS information. With the inputted precursor ion list, Met4DX enables to perform 4D peak detection and related data processing for different IM-MS data including PASEF-DDA and PASEF-DIA data from Bruker TIMS, and IM-AIF data from Agilent DTIM-MS. For example, in NIST human urine samples acquired by PASEF-DDA, Met4DX detected 11419 features in total. Among them, 8309 features (73%) had MS2 spectra (**Supplementary Figure 28a, Supplementary Data 10**). As a comparison, MS-DIAL and

MetaboScape detected 14440 and 4351 features, and 3741 (26%) and 3927 (90%) features had MS2 spectra, respectively. The percentage of 4D features with MS2 spectra in MS-DIAL was significantly lower than Met4DX and MetaboScape. Moreover, Met4DX also showed higher quantification precision with a median RSD of 16.5%. The percentage of detected peaks with RSD less than 30% was as high as 75% in Met4DX (**Supplementary Figure 28b**). As a sharp contrast, although MS-DIAL generated more 4D features, most of them had poor quantification precision (median RSD=47.8%; **Supplementary Figure 28c**). Moreover, we overlapped 4D features detected by Met4DX and MS-DIAL (**Supplementary Figure 29a**). As a result, 5263 features were obtained from both software tools, while 6156 and 9711 features were obtained only from Met4DX and MS-DIAL, respectively. However, these additional features obtained from MS-DIAL had low quantification precision with a median RSD of 56.3% and percentage of peaks with RSD less than 30% was as low as 11% (**Supplementary Figure 29b**). In addition, only 11% of 4D peaks had MS2 spectra acquired (**Supplementary Figure 29d**). These results indicated that additional peaks detected by MS-DIAL and missed by Met4DX had low quality. As a sharp comparison, additional features from Met4DX showed much higher quantification precision and MS2 coverage (**Supplementary Figure 29c and e**).

We also demonstrated the capability of Met4DX to process PASEF-DIA data from Bruker TIMS (**Supplementary Figure 30, Supplementary Data 11**) and AIF data from Agilent DTIM-MS instruments (**Supplementary Figure 31, Supplementary Data 12**). In NIST human urine data acquired by PASEF-DIA from TIMS, Met4DX detected 10021 features. Among them, 7882 features (79%) had MS2 spectra extracted (**Supplementary Figure 30a**). As a comparison, for the same data set, MS-DIAL detected 10681 features, but only 104 features had MS2 spectra extracted (<1%, **Supplementary Figure 30b**). MetaboScape detected 5753 features and did not support MS2 spectral extraction for DIA data (**Supplementary Figure 30c**). Together, these results demonstrated that Met4DX enabled high-coverage 4D peak detection and MS2 spectral extraction in PASEF-DIA data compared with other software tools. Both MS-DIAL and MetaboScape have to improve MS2 spectral extraction function for PASEF-DIA data. For IM-AIF data from Agilent DTIM-MS, Met4DX detected 8324 features in NIST human urine data, and 8318 features had MS2 spectra extracted (**Supplementary Figure 31**). Similarly, MS-DIAL enabled to detect 9054 features and 9042 features had MS2 spectra extracted. This indicated both Met4DX and MS-DIAL enabled to process IM-AIF data from Agilent instrument. Altogether, we have demonstrated that Met4DX is a versatile data processing tool for various 4D metabolomics data with high-coverage and quantification precision in peak detection. ”.

Comment #2: “figure 2e - what is the open circle/closed circle represent?”

Ans: Thanks a lot for the reviewer’s comment. These symbols were used to represent the scan mode in mass spectrometric analyses, which were originally defined by R. G. Cooks and co-workers (Systematic delineation of scan modes in multidimensional mass spectrometry; *Anal. Chem.*, 1990, 62, 1809–1818; doi.org/10.1021/ac00216a016). Specifically, in **Figure 2e**, the closed circle represents a fixed mass while the open circle represents the variable mass. The arrow indicates the mass transition. These combined symbols represent a product ion scan to generate an MS2 spectrum from a fixed precursor ion. In revised manuscript, we have added the related description in the figure caption.

Comment #3: “2-3 fold more peaks than MS-Dial or Metaboscape. How do we know these are real signals? would adjusting the sensitivity settings in MSDial (for example, the 300 minimum amplitude setting) enable higher feature counts? does the fidelity drop for those peaks that Met4Dx detects but are missed by MSDial, for example? what does 'manually checked' mean. looking at 7000+ manual 4D integration peaks per injection? I doubt that can be done well, and it is probably not what you actually did.”

Ans: Thanks a lot for the reviewer’s comment. As suggested by the reviewer, we set the minimum peak height to 0 amplitude for peak detection in MS-DIAL. The number of 4D peaks of NIST human urine data in positive mode increased from 14440 to 15659 (8% increase). Among them, the number of 4D peaks with MS2 spectra increased from 3741 to 3811 (2% increase). These changes were minor (**Figure R1**).

Figure R1. Numbers of 4D peaks detected in NIST human urine samples (PASEF-DDA; positive mode) using MS-DIAL with different settings of minimum peak heights (300 vs 0 amplitude).

Figure R2. Peak fidelity rates of overlapped 4D peaks between Met4DX and MS-DIAL and 4D peaks only detected by Met4DX in NIST human urine samples (PASEF-DDA; positive mode). Manual check was performed here to calculate peak fidelity rates.

To evaluate whether the additional 4D peaks detected by Met4DX but missed by MS-DIAL were true signals, we manually checked the fidelity of these additional peaks (N = 4614), and compared with the fidelity rate of overlapped peaks detected by both Me4DX and MS-DIAL (N = 2673) in NIST human urine data (positive mode). Peak fidelity rates were 99% and 95% for overlapped peaks and peaks only detected in Met4DX, respectively (**Figure R2**). Therefore, we believe the additional peak detected by Met4DX were mostly true signals. Specific examples in **Figure 5f** (detected by Met4DX but missed by MS-DIAL and MetaboScape) also validated this conclusion. Additionally, for Met4DX, all detected 4D peaks have acquired MS2 spectra triggered by PASEF-DDA. In DDA, MS2 spectral acquisition was chosen and ranked by the intensities of precursor ions. Thus, higher abundant ions tend to have MS2 spectra acquired. In principle, they had higher probabilities to be true signals. Taken together, considering the results from manual check and basic principle of PASEF-DDA, we believe the additional peak detected by Met4DX were mostly true signals.

To clarify and validate the peak fidelity rates obtained from manual check, the following description and discussion was added in the revised manuscript. First, to manually check whether a detected 4D peak is a true signal, we exported EIC and EIM peak plots of each detected 4D peak. In practice, we only selected the replicate sample with the highest peak intensity for evaluation. For NIST human urine sample, we exported and manually checked 7287 EIC and 7287 EIM peaks in positive mode, and 6889 EIC and 6889 EIM peaks in negative mode, respectively. In manual check, a 4D peak with good EIC or EIM peak shape was recognized as a true 4D peak. As a result, 97% and 93% of 4D peaks were manually checked with good peak fidelity in positive and negative modes, respectively (**Figure 3g**). Indeed, the manual check took a huge amount of time and effort. Second, we also employed the bioinformatic tool EVA to evaluate the peak fidelity. For EVA, the same EIC and EIM peak plots were used for evaluation. EVA also reported high peak fidelity rates of 96% and 95% in positive and negative modes, respectively (**Figure 3g**). In revised manuscript, we also validated the consistency of peak fidelity evaluations between manual check and EVA (**Supplementary Figure 7**). As a result, 93% and 89% of peak fidelity evaluations were consistency between manual check and EVA in positive and negative modes, respectively. These results ensure the validity of peak fidelity results from both manual check and EVA.

Supplementary Figure 7. Consistency of peak fidelity results obtained from manual check and bioinformatic tool EVA in NIST human urine samples (PASEF-DDA; positive and negative modes).

In revised manuscript, we have added the following results and descriptions:

“We selected the replicate sample with the highest peak intensity for evaluating peak fidelity. For NIST human urine samples, we exported and manually checked 7287 EIC and 7287 EIM peaks in positive mode, and 6889 EIC and 6889 EIM peaks in negative mode, respectively. For NIST human urine samples, 97% and 93% of 4D peaks were manually checked with good peak fidelity in positive and negative modes, respectively (Figure 3g). Also, EVA checked the same samples and reported the similar peak fidelity rates of 96% and 95% in positive and negative mode, respectively. In addition, 93% and 89% of peak fidelity evaluations were consistency between manual check and EVA in positive and negative modes, respectively (Supplementary Figure 7).”

Comment #4: *“Figure 5b - what do the three bars represent? no legend for colors.”*

Ans: Thanks a lot for the reviewer’s comment. The bars and colors in **Figure 5b** represent different confidence levels of metabolite annotation. We have added the figure legend to **Figure 5b** in the revised manuscript.

Comment #5: *“Figure 5c - what does the N= 16M refer to? Is these isomeric feature pairs? Why would you scale the IMS CCS values (4, 2, 0.5%) but not the retention time values? Clearly with better chromatographic resolution you would also improve your identification statistics. Further, IMS has poor orthogonality to m/z relative to reverse phase LC.”*

Ans: Thanks a lot for the reviewer’s comment. In Met4DX, we collected the metabolites in KEGG and HMDB, and generated a library containing 135,638 metabolites. Among them, 125,163 metabolites have at least one isomeric metabolite (metabolite with the same formula), which generated a total of 16,484,071 pairs of isomeric metabolite pairs (**Figure 5c**). We have revised the related caption in the revised manuscript to clarify this.

As suggested by the reviewer, we also evaluated how LC influenced the separation of isomeric metabolites (**Supplementary Figure 14**). For all metabolites in the library, we predicted their RTs in both HILIC and RP conditions using the neural network-based approach (see details in **Methods**; Page 26-27). The RT perdition errors were 17.62 and 9.65 s in HILIC and RP separations, respectively (**Supplementary Figure 32**). Experimentally, the peak width is roughly 10 s in our LC separation. Thus, we set the RT differences (Δ RT) as 20 s, 10 s and 5 s, respectively, to evaluate how LC influenced the separation of isomeric metabolites. As seen in **Supplementary Figure 14b**, when Δ RT were set as 20 s, 10 s and 5 s, roughly 2%, 18%, and 56% of isomeric metabolites were differentiated in HILIC separation, respectively. The similar results were obtained for RPLC mode (**Supplementary Figure 14d**). As a comparison, for same isomeric metabolites, when Δ CCS were set as 4%, 2% and 0.5%, 34% and 92% of isomeric metabolites were differentiated, respectively (**Supplementary Figure 14a**). Additionally, the combination of LC and IM separation further improved the differentiation of isomeric metabolites. As a result, 36%, 62% and 93% of isomeric metabolites were differentiated with LC and IM dual separation with different combinations of Δ RT and Δ CCS, respectively (**Supplementary Figure 14c and e**).

Clearly, IM separation had higher capability than LC to separate isomeric metabolites in theory. We think LC separation showed relatively poor performances because either HILIC or RP mode separates partial isomeric metabolites according to their polarities while IM is general separation regardless of polarities. In addition, most of RTs were predicted in our library using the neural network-based approach. LC separation power might be under-estimated due to the low prediction accuracy of RT compared with relatively high prediction accuracy of CCS values (~1-2%). Therefore, in the **Figure 5c**, we emphasized the IM separation instead of LC separation. We have added the relative results to the revised manuscript as suggested by the reviewer in the part of “Metabolite annotation and isomer differentiation”.

Supplementary Figure 14. Differentiation of isomeric metabolites with the same formula in 4D metabolite library (N=16,484,071 pairs) with IM separation (a), LC separation (b for HILIC and d for RPLC) and LC×IM dual separations (c for HILIC×IM and e for RPLC×IM).

Comment #6: “Figure 5d - it would be useful to show some examples where MSDial/Metaboscape failed to pick these up. (maybe this is Supplementary Figure 17)??”

Ans: Thanks a lot for the reviewer’s comment. The examples in **Figures 5f** and **Supplementary Figure 17** (the figure is re-number as **Supplementary Figure 21** in the revised version) were the co-eluted isobaric feature pairs detected by Met4DX but missed by MS-DIAL and MetaboScape. We have revised the related caption in the revised manuscript to clarify this.

Comment #7: “Figure 5e - what does it look like if you do the same for retention time? ie. ppm < 10 ppm, but IMS values are the same with different retention times?”

Ans: Thanks a lot for the reviewer’s comment. As suggested by the reviewer, we also evaluated the RT differences for isobaric feature pairs with $\Delta m/z \leq 10$ ppm and $\Delta CCS \leq 2\%$ as IM co-eluting metabolite pairs (**Supplementary Figure 23**). There were 3691 features with at least one IM co-eluting isobaric feature, generating a total of 14528 IM co-eluting isobaric feature pairs. Among them, 12690 pairs (87%) had ΔRT larger than 20 s. In addition, 949, 406 and 483 pairs had ΔRT in the ranges of 10-20 s, 5-10 s, and 0-5 s, respectively. This result demonstrated the importance of LC separation in multidimensional metabolomics using LC–IM–MS. We have added the relative result to the manuscript as suggested by the reviewer in the part of “Metabolite annotation and isomer differentiation”.

Supplementary Figure 23. The distributions of ΔRT for isobaric feature pairs in multidimensional metabolomic analysis using LC–IM–MS. Isobaric feature pairs with $\Delta m/z \leq 10$ ppm and $\Delta CCS \leq 2\%$ were used for statistics here. The data was from mouse liver tissue samples.

Comment #8: “There is a fair amount of language claiming that IMS is great for isomer separation in a manner that chromatography isn’t. I do not see data supporting this perspective in this paper, and published literature suggests that IMS is less orthogonal to mass than retention time. IMS does show a great deal of potential, but the presentation seems unrealistically optimistic to me. Increasing your chromatographic resolution also improves isomer separation! IMS is faster than chromatography but also occurs postionization, meaning that chromatography is necessary to reduce ionization suppression. Overselling IMS doesn’t help. For example, figure 17h demonstrates an example 1% ims separation - most analytical chemists would not consider these to be resolved. Maybe a 2D blot (rt and ims) would help visualize this? 17g represents a 2-3% separation - at first glance it appears the chromatography may actually offer > 2-3% separation - why highlight IMS here?”

Ans: Thanks a lot for the reviewer’s comment. We strongly agree with the reviewer’s comment that LC separation also plays a vital role in multidimensional metabolomic analyses empowered by LC–IM–MS. To strengthen this point, in the revised manuscript, we have added the following statements to highlight the importance of LC separation:

“In multidimensional metabolomic analyses using LC–IM–MS, LC separation also played a vital role in separation of isomeric metabolites in real biological samples. For example, in the mouse liver samples, there were 3691 features with at least one IM co-eluting isobaric feature ($\Delta m/z \leq 10$ ppm and $\Delta CCS \leq 2\%$), generating a total of 14528 IM co-eluting isobaric feature pairs. Among them, 12690 pairs (87%) had ΔRT larger than 20 s (**Supplementary Figure 23**). These results proved the importance of LC separation in multidimensional metabolomics using LC–IM–MS.”.

Supplementary Figure 20. LC×IM dual separation in LC–IM–MS-based metabolomics provided better resolving power for co-eluting isobaric feature pairs.

We have followed the suggestion from the reviewer. In **Supplementary Figure 20**, we have provided the 2D plot for examples in **Supplementary Figure 17g and 17h** (re-numbered as **Supplementary Figure 23g and h** in the revised manuscript). They also demonstrated the importance of LC×IM dual separation in LC–IM–MS-based metabolomics. The related description has been added to the revised manuscript as follows:

“Met4DX also enabled to discriminate co-eluted pairs of isobaric features with ΔCCS as small as 1–2% (**Figure 5i and Supplementary Figure 19**), which could only be partially separated by IM and better resolved with LC×IM dual separation in LC–IM–MS-based metabolomics (**Supplementary Figure 20**).”.

Comment #9: “Computational time?”

Ans: Thanks a lot for the reviewer’s comment. To calculating the computational time, we run these software tools with NIST human urine data (positive mode) on a desktop with Intel Core i7-12700 (2.10 GHz; 12 CPU cores; 20 logical cores) and 32 GB memory. For computational time, MS-DIAL, MetaboScape and Met4DX took **3, 6 and 20** minutes to finish the data processing. For memory usage, MS-DIAL, MetaboScape and Met4DX occupied ~4-6 GB, ~1-1.5 GB and ~6-9 GB, respectively. In general, the computing resource required by Met4DX is affordable for most users.

The related description has been added to the revised manuscript as follows: “For computational resources comparison, we run these software tools with NIST human urine data (positive mode) on a

desktop with Intel Core i7-12700 (2.10 GHz; 12 CPU cores; 20 logical cores) and 32 GB memory. For computational time, MS-DIAL, MetaboScape and Met4DX took 3, 6 and 20 minutes to finish the data processing. For memory usage, MS-DIAL, MetaboScape and Met4DX occupied ~4-6 GB, ~1-1.5 GB and ~6-9 GB, respectively. In general, the computing resource required by Met4DX is affordable for most users.”.

Comment #10: *“EVA appears from the publication to be trained to estimate quality for 3D (LC-MS-Intensity) peaks. Does it work for 4D data, or did you need to remove a dimension to enable this evaluation?”*

Ans: Thanks a lot for the reviewer’s comment. The review is correct that EVA was originally trained to evaluate peak fidelity for 3D peaks. The extracted ion chromatogram (EIC) plots were used as input. To make EVA applicable for 4D peak, we generated both EIC and EIM peak plots for each 4D peak. Then, one 4D peak had two 3D peaks. This is the exactly same as the reviewer mentioned. For EIC peaks, the IM dimension was removed, while for EIM peaks, the LC dimension was removed. Then, both EIC and EIM peak plots for each 4D peak were inputted into EVA for peak fidelity evaluation. In our analysis, a 4D feature with good EIC or EIM peak shape was recognized as a true 4D peak. We have revised the related methods in the revised manuscript to clarify this.

Reviewer #2:

“The manuscript with title “A mass spectrum-oriented computational method for ion mobility-resolved untargeted metabolomics” presents Met4DX, an end-to-end computational framework which performs peak detection, quantification, and identification of metabolites in IM-resolved 4D metabolomics with data-dependent acquisition. The peak detection strategy represents a great alternative to existing tools. I fully agree that more informatic solutions are needed to promote widespread application of 4D untargeted metabolomics. The software was implemented in R, the source code was provided with an accompanying container and instructions to use it. The performance of Met4DX was benchmarked against other existing tools using different standard complex biological samples. The manuscript is overall well written, and the results well presented.”

Ans: Thanks a lot for the reviewer’s positive comments towards publication. We feel your constructive comments significantly strengthen the manuscript considerably.

“The following points should be addressed to improve the clarity of the work:”

Comment #1: *“The Abstract and Introduction should specify that the workflow is for data-dependent acquisition: IM-resolved 4D data-dependent acquisition metabolomics.”*

Ans: Thanks a lot for the reviewer’s comment. In the revised manuscript, we followed Reviewer #1’s suggestion, and modified Met4DX to support the input of **a list of precursor ions** as “seeds” to initiate 4D peak detection. With the inputted precursor ion list, Met4DX enables to perform 4D peak detection and data processing for different IM-MS data including **PASEF-DDA and PASEF-DIA data from Bruker TIMS, and IM-AIF data from Agilent DTIM-MS** (see **Supplementary Figure 26-31** and the response to Comment #1 of Reviewer #1 for details). With these new functions in Met4DX, we feel we should not revise the related statements in the abstract and introduction.

2. Met4DX Workflow.

Comment #2a: *“How is defined the intensity in the MS2 spectrum with the highest spectral intensity to be selected to represent the cluster? The sum of all MS2 peaks?”*

Ans: Thanks a lot for the reviewer’s comment. The spectral intensity in an MS2 spectrum is the sum intensity of top 10 fragment ions ranked by their intensities. In the revised manuscript, the definition of MS2 spectral intensity were add to the **Methods**: *“The spectral intensity in an MS2 spectrum is the sum intensity of top 10 fragment ions ranked by their intensities.”*

Comment #2b: *“Limitation, Met4DX requires MS2, how many features could be detected by MS-DIAL and missed by Met4DX? Discuss this limitation of the Met4DX workflow. While the Met4DX approach provides advantages, many metabolites could be detected without any MS2 (either due to poor*

fragmentation of the specific molecule or to low concentration), but still the annotation based on m/z , RT and CCS could provide biological insights.”

Ans: Thanks a lot for the reviewer’s comment. Following both reviewers’ suggestions, in the revised manuscript, we modified Met4DX to support the input of a **list of precursor ions** as “seeds” to initiate 4D peak detection (**Supplementary Figure 26 and 27**). In revised Met4DX, we have also curated a list of precursor ions collected from various biological samples (N=72,265 in positive mode; N= 42,553 in negative mode; **Supplementary Data 9**). Each ion includes m/z , RT, and CCS information. With the inputted precursor ion list, Met4DX enables to perform 4D peak detection without MS2 spectra.

Supplementary Figure 28. High-coverage and high quantification precision of 4D peak detection in Met4DX using the inputted precursor ion list for peak detection in NIST human urine data (PASEF-DDA; positive mode; n=6 technical replicates).

Supplementary Figure 29. 4D peaks obtained from Met4DX showed higher quantification quality and MS2 spectral coverage than those from MS-DIAL. (a) Overlaps of 4D peaks obtained from Met4DX and MS-DIAL. (b-c) The distributions of relative standard deviation (RSD) of 4D peaks from MS-DIAL (b) and Met4DX (c). (d-e) The MS2 spectral coverage of 4D peaks obtained from MS-DIAL (d) and Met4DX (e). NIST human urine data (PASEF-DDA; positive mode; n=6 technical replicates) was used here.

For example, with the list of precursor ions, in NIST human urine samples (positive mode), Met4DX enabled to detect 11419 4D features in total. Among them, 8309 features (73%) had MS2 spectra (**Supplementary Figure 28a**). As a comparison, MS-DIAL and MetaboScape detected 14440 and 4351 features, and 3741 (26%) and 3927 (90%) features had MS2 spectra, respectively. The percentage of 4D features with MS2 spectra in MS-DIAL was significantly lower than Met4DX and MetaboScape. Moreover, Met4DX also showed higher quantification precision with a median RSD of 16.5%. The percentage of detected peaks with RSD less 30% was as high as 75% in Met4DX (**Supplementary Figure 28b**). As a sharp contrast, although MS-DIAL generated more 4D features, most of them had poor quantification precision (median RSD=47.8%; **Supplementary Figure 28c**).

In addition, to evaluate the consistency of peak detection between Met4DX and MS-DIAL, we overlapped 4D features detected by Met4DX and MS-DIAL (**Supplementary Figure 29**). As a result, 5263 features were obtained from both software tools, while 6156 and 9711 features were obtained from Met4DX and MS-DIAL, respectively (**Supplementary Figure 29a**). However, we further demonstrated that these additional features obtained from MS-DIAL had low quantification precision with a median RSD of 56.3% and percentage of peaks with RSD less than 30% was as low as 11% (**Supplementary Figure 29b**). In addition, only 11% of 4D peaks had MS2 spectra acquired (**Supplementary Figure 29d**). These results indicated that additional peaks detected by MS-DIAL and missed by Met4DX had low quality. As a sharp comparison, additional features from Met4DX showed much higher quantitation precision and MS2 coverage.

We have added these relative results and statements in the revised manuscript:

*“Moreover, we overlapped 4D features detected by Met4DX and MS-DIAL (**Supplementary Figure 29a**). As a result, 5263 features were obtained from both software tools, while 6156 and 9711 features were obtained only from Met4DX and MS-DIAL, respectively. However, these additional features obtained from MS-DIAL had low quantification precision with a median RSD of 56.3% and percentage of peaks with RSD less than 30% was as low as 11% (**Supplementary Figure 29b**). In addition, only 11% of 4D peaks had MS2 spectra acquired (**Supplementary Figure 29d**). These results indicated that additional peaks detected by MS-DIAL and missed by Met4DX had low quality. As a sharp comparison, additional features from Met4DX showed much higher quantitation precision and MS2 coverage (**Supplementary Figure 29c and e**).”*

Comment #3: *“Figure 1-b. In the first panel, what do the equal sign and the dot mean? Add next to it the text “Precursor m/z, RT, CCS” for clarity.”*

Ans: Thanks a lot for the reviewer’s comment. The dot means the precursor ion of an MS2 spectrum. We have revised the **Figure 1b** to clarify the workflow. The text of “m/z, RT, mobility” and the legend for black dot were added into the revised figure.

Revised Figure 1b in the manuscript main text.

4. Performance benchmark.

Comment #4a: “MetaboScape. Indicate the first time it is mentioned and in the Methods that this is a proprietary software from Bruker.”

Ans: Thanks a lot for the reviewer’s comment. We have added the related statement in the revised manuscript as follows: “Other software tools such as MS-DIAL and MetaboScape (Bruker Daltonics, Bremen, Germany) were used for comparison” in the main text, and “For MetaboScape (Bruker Daltonics, Bremen, Germany, version 2022b), raw data files were imported for peak detection” in the **Methods**.

Comment #4b: “Can you add a few sentences about benchmarking performance regarding computing resources (run time and memory used)?”

Ans: Thanks a lot for the reviewer’s comment. To calculating the computational time, we run these software tools with NIST human urine data (positive mode) on a desktop with Intel Core i7-12700 (2.10 GHz; 12 CPU cores; 20 logical cores) and 32 GB memory. For computational time, MS-DIAL, MetaboScape and Met4DX took 3, 6 and 20 minutes to finish the data processing. For memory usage, MS-DIAL, MetaboScape and Met4DX occupied ~4-6 GB, ~1-1.5 GB and ~6-9 GB, respectively. In general, the computing resource required by Met4DX is affordable for most users.

The related description has been added to the revised manuscript as follows: “For computational resources comparison, we run these software tools with NIST human urine data (positive mode) on a desktop with Intel Core i7-12700 (2.10 GHz; 12 CPU cores; 20 logical cores) and 32 GB memory. For computational time, MS-DIAL, MetaboScape and Met4DX took 3, 6 and 20 minutes to finish the data processing. For memory usage, MS-DIAL, MetaboScape and Met4DX occupied ~4-6 GB, ~1-1.5 GB and ~6-9 GB, respectively. In general, the computing resource required by Met4DX is affordable for most users.”

Comment #5: “It seems the tool used to generate the required MGF files, *DataAnalysis*, is also a Bruker software, this could potentially hinder the utilization. If it is a proprietary software, can you provide an alternative using free/open-source software? Clarify this point in the manuscript.”

Ans: We agree with the reviewer’s comment. In the revised manuscript, we followed the reviewer’s great suggestion and added a new function named “GenerateMS2”, which enables to generate MS2 spectra directly from the raw data. **Thus, *DataAnalysis* software from Bruker is not required for Met4DX.** To further validate the function of “GenerateMS2”, we compared the peak detection consistency using MS2 spectra generated from Bruker *DataAnalysis* (DA converted) and “GenerateMS2” function in Met4DX (**Supplementary Figure 34**). Their results were highly consistent in both positive and negative modes.

The related description has been added to **Methods** in the revised manuscript as follows:

*“MS2 spectra in each raw MS file were first converted into the .mgf files using *DataAnalysis* (Bruker Daltonics, Bremen, Germany; version 5.2). The raw MS data files and MS2 spectra in mgf files were organized into one folder and imported into Met4DX for data processing. If MS2 spectral files were not provided, Met4DX enabled to generate MS2 spectra directly from the raw data files using the “GenerateMS2” function. Our data showed that Met4DX showed high consistency in peak detection using MS2 spectra converted from *DataAnalysis* and generated by the “GenerateMS2” function (**Supplementary Figure 34**). The demo code to use the “GeneratedMS2” function was provided in GitHub (<https://github.com/ZhuMetLab/Met4DX>).”*

Supplementary Figure 34. Met4DX showed high consistency in peak detection using MS2 spectra converted from Bruker *DataAnalysis* software (DA converted) and generated by the “GenerateMS2” function in Met4DX. The data was acquired from NIST human urine samples with PASEF-DDA in positive mode (n=6 technical replicates).

Comment #6: “Quantification precision. Perform an evaluation of the quantitation linearity using the dilution data (serially diluted urine samples). This will better demonstrate the accuracy of the peak integration in Met4DX, beyond number of IDs and comparison by pair of replicates. For example, see *DIAMetAlyzer* 10.1038/s41467-022-29006-z, Figure 3-c.”

Ans: Thanks a lot for the reviewer’s comment. In serially diluted urine samples, the matrix effects in different diluted samples were quite different and significantly influenced electrospray ionization, which

made the metabolite quantitation linearity poor (data not shown). Alternatively, we prepared a mixture of 20 natural products (not contained in urine sample) and spiked it into NIST human urine samples in a 4-fold dilution series and evaluated the quantitation linearity of Met4DX. All of them were successfully measured by LC-IM-MS and detected by Met4DX. Each sample was acquired with 6 technical replicates. The median peak areas of each natural product under one dilution condition were normalized by the median peak area of its highest concentration (i.e., 1X dilution). As seen in **Supplementary Figure 11**, Met4DX obtained good quantitation linearity for serial diluted samples. Please note that intensities of most natural products beyond 256-fold dilution were too low to be detected by TIMS instrument. Therefore, we set our serial dilution experiment from 1-fold to 256-fold.

Supplementary Figure 11. Normalized intensity ratio over the dilution series of 20 natural products in NIST human urine samples. Each sample was acquired with 6 technical replicates and the median peak areas of each natural product under one dilution condition were normalized by the median peak area of its highest concentration. The dashed line indicates the expected four-fold difference along the dilution.

In the revised manuscript, the related description has been added as follows: “Furthermore, we evaluated the quantitation linearity of Met4DX. A mixture of 20 natural products was spiked into NIST human urine samples in a 4-fold dilution series. All of natural products measured by LC-IM-MS and detected by Met4DX showed good quantitation linearity (see **Methods** and **Supplementary Figure 11; Supplementary Data 7**)”.

Comment #7: “The methods indicate that a library containing predicted values of CCS, RT and MS/MS was generated for annotation. This could be a valuable resource for the community. Is it included in the container? If not, please share the library and specify in the manuscript and instructions how to use it. Furthermore, specify how the users can utilize their own library in Met4DX.”

Ans: Thanks a lot for the reviewer’s comment. In original version of Met4DX, the full version of the metabolite library was included in the container and could be used for metabolite annotation. The demo code was provided in GitHub (<https://github.com/ZhuMetLab/Met4DX>).

In addition, we added **Supplementary Note 1** in the revised Supplementary Information to specify how user can utilize a user-defined library. The related description has been added to **Methods** as follows: “Met4DX also supported metabolite annotation with a user-defined library, and the instruction was provided in the **Supplementary Note 1**.”

Comment #8: “Figure 5-b: indicate in a better way the confidence levels. For example, if the levels are the colors related to figure 5-a, add a rectangle joining a and b, or indicate it in the figure legend. Furthermore, consider including Supplementary Figure 10 as part of figure 5.”

Ans: Thanks a lot for the reviewer’s comment. We have added the figure legend to **Figure 5b** suggested by the reviewer.

Comment #9: “Figure 5-d. What was the CCS tolerance to evaluate the number of co-eluted isobaric pair features detected by the different software tools?”

Ans: Thanks a lot for the reviewer’s comment. Here, we defined the feature pair with $\Delta m/z \leq 10$ ppm and $\Delta RT \leq 10$ s as the co-eluted isobaric feature pairs in each software tools, which means no CCS tolerance was applied to search co-eluted isobaric feature pairs. We have revised the figure caption to clarify this.

Comment #10: “Related to CCS tolerance, the Methods indicate that the minimum and maximum tolerances of trapezoidal function for CCS match were set as 3% and 6%, but Figure 5-c compares Δ CCS set as $\geq 4\%$, $\geq 2\%$, and $\geq 0.5\%$. Clarify these differences and how ambiguity is resolved when different annotations are matched to the same feature, or the same metabolite is matched to different features. Also, discuss the limitation of using predicted CCS values which have larger errors than the experimental values that can be obtained in current IM instrumentation.”

Ans: Thanks a lot for the reviewer’s comment. CCS values in the metabolite library were generated by AllCCS developed by our group (<http://doi.org/10.1038/s41467-020-18171-8>). According to the results in AllCCS paper, 72% of predicted CCS values had relative error less than 3% while 93% of predicted CCS values had relative error less than 6% (**Supplementary Figure 33**). Therefore, we set 3% and 6% as the minimum and maximum tolerances of trapezoidal function for CCS match. Meanwhile, the CCS match tolerances were user-defined. If the user inputs a library with CCS values with high accuracy, the tolerances could be reduced.

For clarify, the related description has been added to **Methods** as follows: “The CCS match tolerances were set according to the CCS prediction accuracy of AllCCS. The results showed that 72% of predicted CCS values had relative error less than 3% while 93% of predicted CCS values had relative error less than 6% (**Supplementary Figure 33**). The CCS match tolerances were user-defined. If the user inputs a library with CCS values with high accuracy, the tolerances could be reduced.”

Supplementary Figure 33. Cumulative percentages of predicted CCS values of external validation sets in AllCCS. There were 72% of CCS values with relative errors less than 3% and 93% of CCS values with relative errors less than 6%.

In **Figure 5c**, we performed a simulation analysis using the metabolite library data to demonstrate how IM improves the separation of isomeric metabolite pairs (two metabolites with the same formula and $\Delta RT \leq 10$ s). According to previous publications (Ref 28, 29, and 30), most modern commercial IM-MS instruments had an IM resolving power (ca. 50-100), which enabled to resolve metabolite isomer pairs with a baseline separation when their ΔCCS was larger than 4%, or with a half-peak-width separation when their ΔCCS was larger than 2%. For isomeric pairs with 0.5% of ΔCCS , the separation can only be achieved when the IM resolving power reaches >250 . Therefore, we compared ΔCCS set as $\geq 4\%$, $\geq 2\%$, and $\geq 0.5\%$ in **Figure 5c**. We have revised the related statement to clarify this.

To demonstrate how ambiguity was resolved in annotation of metabolite isomer pairs, we took examples in **Figure 5j** to illustrate the annotation details in **Supplementary Figure 24**. Met4DX performed m/z , RT, CCS and MS2 spectral matches sequentially. After the m/z and RT matches, the co-eluting isomeric metabolites (N-acetyl-L-phenylalanine and 3-phenylpropionylglycine) both matched the co-eluted isobaric feature pair of M206T168C147 and M206T166C155 with high RT match scores. The differences in CCS matches generated distinct CCS match scores and further differentiate the metabolite isomers. Finally, combined with MS2 spectral match scores, M206T168C147 and M206T166C155 were annotated as N-acetyl-L-phenylalanine and 3-phenylpropionylglycine, respectively. We have added the related results and descriptions in the revised manuscript as following:

*“The co-eluted isobaric feature pairs were annotated as the isomeric metabolites of N-acetyl-L-phenylalanine and 3-phenylpropionylglycine, respectively, with the sequential m/z , RT, CCS and MS2 spectral matches. Among them, the CCS match successfully differentiated them (**Supplementary Figure 24**).”*

	N-acetyl-L-phenylalanine_[M-H]- m/z = 206.0822 Da; RT = 171 s; CCS = 146.7 Å ²			3-phenylpropionylglycine_[M-H]- m/z = 206.0822 Da; RT = 170 s; CCS = 155.0 Å ²		
M206T168C147 m/z = 206.0828 Da RT = 168 s CCS = 147.0 Å ²	RT error	CCS error	MS2 score	RT error	CCS error	MS2 score
	3s	0.2%	0.83	2s	5.1%	0.05
	RT score	CCS score	Final score	RT score	CCS score	Final score
	1	1	0.93 ✓	1	0.28	0.33 ✗
M206T166C155 m/z = 206.0826 Da RT = 166 s CCS = 155.3 Å ²	RT error	CCS error	MS2 score	RT error	CCS error	MS2 score
	5s	5.8%	0	4s	0.2%	0.99
	RT score	CCS score	Final score	RT score	CCS score	Final score
	1	0.05	0.22 ✗	1	1	0.99 ✓

Supplementary Figure 24. Metabolite annotations of co-eluted isobaric features M206T168C147 and M206T166C155.

For the limitations of predicted CCS values in AllCCS, the related description has been added to **Discussion** as follows:

“For 4D metabolomics, in the past decade, many CCS prediction tools with high accuracy were developed (with CCS prediction errors of 1-2%). However, the CCS prediction accuracy highly depended on the coverage of training datasets and training models. For some compounds lacking similar structures in training datasets, these tools generated predicted CCS values with relatively large errors, introducing false positives/negatives during metabolite annotation. Alternatively, the curation of experimental CCS values from chemical standards is promising to achieve better performances in metabolite annotation.”.

Comment #11: *“Related to limitations, discuss the importance and future plans to include an error rate estimation method (i.e., false discovery rate) and a graphical-user interface (user-friendly tools are paramount to truly promote widespread application of 4D untargeted metabolomics).”*

Ans: Thanks a lot for the reviewer’s comment. We agreed that both of error rate estimation method for metabolite annotation and graphical-user interface are important for future developments. We have added a paragraph in Discussion to highlight these suggestions:

“Compared with proteomics, error rate estimation (i.e., false discovery rate) is under development in metabolomics. Recently, DIAMetAlyzer reported an automated false-discovery rate-controlled analysis for data-independent acquisition in metabolomics. More specifically for LC–IM–MS based lipidomics, MS-DIAL reported the FDR of lipid annotations during RT and CCS matches using their validation set. Sterol4DAnalyzer developed by our group also reported the FDR of sterols under different CCS match tolerances. A comprehensive FDR estimation for 4D metabolomics annotation should include a ground-truth benchmark dataset and evaluations of all 4-dimensional matches with different tolerances, which would be an important future plan for 4D metabolomics.”.

“Metabolomics software tools with broad utilization are usually developed with user-friendly graphical user interface (GUI) and support all-in-one solution from raw data importing to metabolite annotation, like MS-DIAL, MZmine, XCMS Online and so on. A well-designed GUI enables users to adjust parameters easily and check the result with convenience. The current version of Met4DX could only be run as an R package using command lines. In the future, a GUI with comprehensive functions would make Met4DX more user-friendly.”

Comment #12: *“What are the columns of the multidimensional match results in the “ScoreCombine.csv” file? Do you output only the combined score from the linear weighting function? Please also output each individual score. Do you output the confidence level indicating the type of reference for each dimension (i.e., experimental or predicted)?”*

Ans: Thanks a lot for the reviewer’s comment. **The explanations of each column in “ScoreCombine.csv” file has been added into the revised Supplementary Data file and help document of Met4DX.** In general, columns of “ScoreCombine.csv” included feature information, structural information of annotated metabolites, match errors in three dimensions (*m/z*, RT, CCS), match scores for RT match, CCS match, and MS2 spectral match, combined score, and the confidence level for metabolite annotation. Therefore, yes, each individual match score and the combined score were all outputted. The definition of confidence levels has considered the match with experimental or predicted values (see **Supplementary Figure 12** and **Methods** for details).

Minor:

Comment #13: *“Check missing articles throughout the text. For example, in the Introduction, add “the”: compositional diversity of “the” metabolome.”*

Ans: Thanks a lot for the reviewer’s correction. These typos have been corrected in the revised manuscript.

Comment #14: *“Discussion. Remove “of” in “and generates of four-dimensional data”. Rephrase the last part in “the coverage and accuracy of annotated metabolites in untargeted metabolomics have remained to be improved”.”*

Ans: Thanks a lot for the reviewer’s correction. These typos have been corrected in the revised manuscript. The indicated statement has been removed in the revised manuscript.

REVIEWERS' COMMENTS

Reviewer #1 (Remarks to the Author):

thank you for the extensive revision.

Reviewer #2 (Remarks to the Author):

The authors have addressed the reviewers' concerns and recommendations.

Response to the reviewers:

We appreciate the reviewers' positive comments toward publication.

Reviewer #1:

"thank you for the extensive revision."

Ans: Thanks a lot for the reviewer's positive comments towards publication.

Reviewer #2:

"The authors have addressed the reviewers' concerns and recommendations."

Ans: Thanks a lot for the reviewer's positive comments towards publication.